# Predicting trajectories of illness using RNA velocity of whole blood

Transcriptomic analyses reveal the status of cells, tissues, or organisms, across states of health and disease. RNA velocity adds a temporal dimension to single cell analyses, predicting future transcriptomic and phenotypic states, based on the current spliced and unspliced mRNA of each cell. We hypothesized that RNA velocity could be adapted to predict future clinical state of individuals with acute and chronic illnesses, using their whole-blood transcriptomes. We developed VeloCD, a method for quantitative prediction of transitions in clinical state from a single time-point RNA sample. This predicts transcriptomic trajectories and future infection status in influenza A and SARS-CoV-2 controlled human infection studies, which are consistent with trajectories in naturally acquired infections. In HIV-TB coinfected individuals, VeloCD predicts the onset of immune reconstitution inflammatory syndrome. In individuals receiving biological therapy for inflammatory bowel disease, whole blood RNA velocity after the first dose of treatment indicates whether remission will be achieved by the end of the treatment course. In a multinational observational study of acutely unwell febrile children, VeloCD predicts those with greatest medical care requirements. Our results demonstrate proof-of-concept for the use of RNA velocity to predict trajectories of human diseases.

The blood transcriptome changes in response to illness. Many studies have identified characteristic patterns of gene expression able to distinguish between different diseases or degrees of severity of the same illness[1–11]. Whilst much attention has focused on the diagnostic potential of measuring host gene expression[4], clinicians are also frequently faced with the problem of predicting the progression of acute illness, but prognostic tests based on gene expression have received less attention[12,13].

Most commonly, gene expression analyses utilize mature (spliced) mRNA transcripts, enriched through PolyA selection, or gene-level expression summed across spliced and unspliced (nascent or immature) transcripts. However, recent studies focusing on unspliced transcripts have revealed an additional layer of information about the dynamics of gene expression. These yet-to-be-spliced transcripts, quantified by intron-mapping reads[14], provide an indication of the future spliced transcript expression[15–19]. Taken together, the relative abundance of spliced and unspliced transcripts indicates whether expression of each gene is increasing or decreasing. This concept was formalized through the calculation of RNA velocity, using single-cell RNA-Seq derived measurements of spliced and unspliced transcripts at a single point in time to predict the future transcriptomic state of individual cells and, by reference to the transcriptomes of other cells, to infer future phenotypic states[15]. It was also demonstrated that RNA velocity analysis could be applied to bulk RNA-Seq of mouse liver tissue to model circadian changes in gene expression. Since its original description, RNA velocity methods have been refined and widely applied to understand dynamic and developmental relationships between cellular states across numerous biological systems[20–24].

We reasoned that a practical application of RNA velocity analysis might be its extension beyond projecting the fate of individual cells, adapting the approach to predict the future changes in clinical state of individual people with acute illness, based on a single measurement of

✉e-mail: claire.dunican14@imperial.ac.uk; a.cunnington@imperial.ac.uk

their blood transcriptome. Despite blood having a complex cellular composition, which can change during illness, we and others have demonstrated that it is possible to find transcriptomic signatures in whole blood bulk RNA, which robustly identify different causes and states of acute illness without needing to adjust for cell mixture[25]. We therefore hypothesized that it would be possible to discover RNA velocity-based signatures with whole blood bulk RNA-Seq to make predictions of future transcriptomic and phenotypic states.

To test this hypothesis, we developed an analysis tool: RNA velocity-based transition predictions of future disease states of subjects in clinical datasets (VeloCD). VeloCD makes predictions based on three core assumptions: the current spliced bulk mRNA expression profile in blood indicates current clinical status; the RNA velocity of blood predicts future spliced mRNA expression; future spliced mRNA expression corresponds to future clinical status. To make predictions, RNA velocity is calculated, and future clinical state is defined, with reference to the transcriptomes of individuals with the clinical outcomes of interest, such as presence or absence of a disease phenotype, or mild vs. severe illness. Quantitative probabilities of transition to each reference group are then calculated. Low dimensional fate maps, like those generated for single cell transcriptomics, can then be produced to visualize the evolution of the response to human disease and relationships between different disease states.

Here we show that the core assumptions of VeloCD are valid and provide evidence that it can generate meaningful prognostic information in human diseases. We demonstrate proof-of-concept for predicting future infection after exposure to influenza and SARS-CoV-2 viruses, predicting onset of Tuberculosis (TB)-associated Immune Reconstitution Inflammatory Syndrome (TB-IRIS), predicting response after initiation of biological therapy for Inflammatory Bowel Disease (IBD), and predicting recovery or deterioration in acutely ill febrile children. These findings indicate the potential for developing prognostic tests based on RNA velocity of blood.

## Results

### RNA velocity of blood in controlled human infections

To test the core assumptions underpinning VeloCD, we utilized whole-blood RNA-Seq data (Supplementary Table 1) from two controlled human infection models (CHIMs), in which healthy volunteers were intentionally inoculated with either influenza A virus[26] or SARS-CoV-2[27] and serial blood samples were collected over the following days (Fig. 1a–d). Although VeloCD does not use serial samples to make predictions, the known timings of inoculation, first-polymerase chain reaction (PCR) positivity, and sequential samples in these studies provide rare ground truth data against which the assumptions and predictions of VeloCD can be assessed.

17 of 23 subjects in the influenza study (Fig. 1b), and 17 of 26 in the SARS-CoV-2 study became infected (Fig. 1d), as determined by PCR testing. Amongst infected volunteers there was some variation in time from inoculation to first positive PCR test, and variation in symptom severity, but none became unwell enough to require medical intervention. In those testing positive, the median day of first PCR-positivity was day 2 post-inoculation in both studies, with a range of 1-4 days in the influenza virus study, and 1-2 days in the SARS-CoV-2 study.

We first investigated whether the current spliced transcript expression in blood is representative of the current infection status in these two cohorts. To do this, we quantified the spliced transcript expression of all multi-exon genes with spliced and unspliced transcript expression, resulting in 21,647 RNA transcripts in the influenza study, and 23,903 RNA transcripts in the SARS-CoV-2 study. When spliced transcript expression was transformed into low-dimensional transcriptomic space using principal component analysis (PCA)[28] (Fig. 1e, f), there was some segregation of samples by both time since inoculation and infection status, with the separation of PCR-positive from PCR-negative subjects becoming more prominent from day 2 for

the influenza study and day 3 for SARS-CoV-2 study respectively. The greatest changes in gene expression appeared to occur on the day after first detection of infection by PCR. We used differential expression analysis (DEA) to identify the genes that distinguish PCR-positive and PCR-negative subjects at these time-points of greatest change in gene expression (there were no differentially expressed genes prior to virus inoculation between subjects who became infected or remained uninfected).

In the influenza study, subjects were subset by the day they first tested positive, and the sample from the day after they first tested positive was used for DEA. We identified 2988 differentially expressed genes (DEGs) in day 2 samples between those who first tested positive on day 1 ($n = 6$) and those who remained PCR-negative throughout ($n = 6$) (Fig. 1g). There were 892 DEGs in day 3 samples between those who first tested positive on day 2 ($n = 9$) and those who remained PCR-negative throughout ($n = 6$) (Fig. 1g).

In the SARS-CoV-2 study there were 406 DEGs on day 3 between PCR-positive subjects ($n = 17$) and those who remained PCR-negative throughout ($n = 9$; Fig. 1h). Only a single time-point was used for the SARS-CoV-2 analysis because the PCR-positive subjects had more uniform gene expression change over time. Using these sets of DEGs for PCA, there was clear segregation of PCR-positive and PCR-negative subjects in low-dimensional transcriptomic space at these time-points in both studies (Fig. 1i-j).

To further elucidate differences between the PCR-positive and PCR-negative subjects for each virus, we used the maSigPro backward selection algorithm to identify spliced transcripts changing differentially over time[29]. This identified 1563 and 2700 genes for influenza and SARS-CoV-2 infections, respectively. Two of the top five genes (ordered by multiple testing corrected $p$ value) were shared across these analyses: Interferon Induced Protein 44-Like gene (*IFI44L*) and Lymphocyte antigen 6E (*LY6E*) (Fig. 1k, l show their spliced transcript expressions over time). PCA using the genes selected by maSigPro (Fig. 1m, n) shows clear temporal ordering of the samples from PCR-positive, infected subjects, which are clearly distinct from PCR-negative subjects. Taken together, these results from two independent CHIMs indicate that whole blood bulk spliced mRNA expression profiles represent current infection status at different timepoints across the course of an acute infection, from exposure through convalescence, discriminating between infected (PCR-positive) and uninfected (PCR-negative) states.

Next, we investigated whether RNA velocities could capture future gene expression and corresponding future disease states (defined by PCR status) in these CHIM datasets. Although RNA velocity does not require serial samples to predict trajectories, it is helpful to validate VeloCD using serial samples to show that it produces accurate prediction of future transcriptomic states. We first assessed whether unspliced transcript expression of the genes selected by maSigPro follows biologically informative patterns across time, including upregulation and repression over the course of infection. Unspliced transcripts of *IFI44L* and *Ly6E* (Fig. 2a–d) showed parallel profiles of expression to their respective spliced transcripts (Fig. 1k, l). These figures also demonstrate that the changes in spliced and unspliced transcript expression are apparent before individuals test positive by PCR, which is also evident in the corresponding phase portraits (Supplementary Fig. 1).

Future spliced transcript expression values are predicted as an intermediate step in the calculation of RNA velocity values[15]. For both CHIM datasets, we used day 2 samples to predict future spliced transcript expression values for all the genes identified as significant in the maSigPro analysis and correlated the predicted future spliced transcript expression with the measured spliced transcript expression on day 3 (Fig. 2e, f). The range of Pearson correlation coefficients across subjects were 0.86-0.99 (mean, 0.96) and 0.77-0.99 (mean, 0.94) in the influenza and SARS-CoV-2 datasets, respectively, indicating that

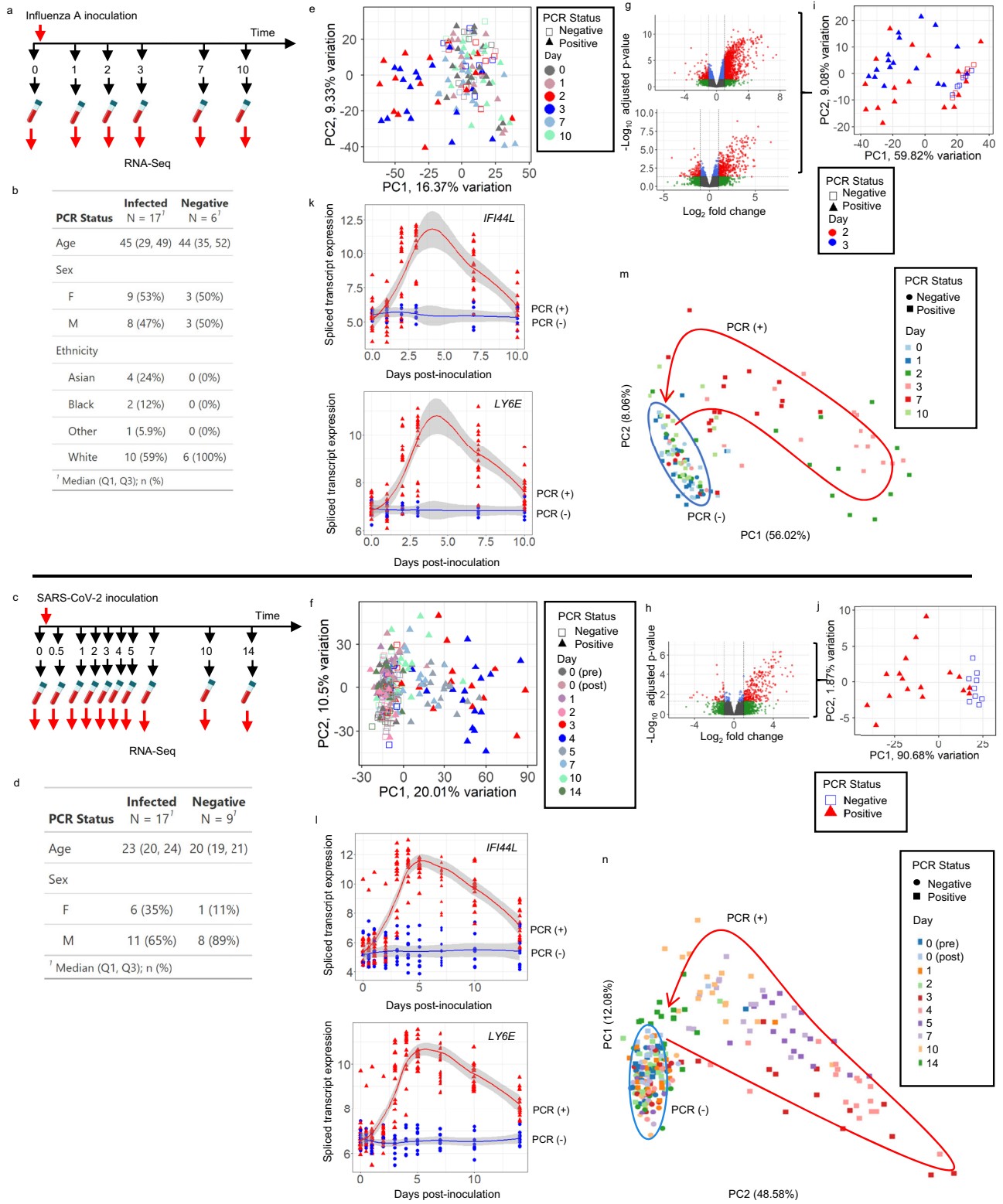

this method accurately models the temporal expression of these genes up to 24 hours in the future.

In the influenza dataset, future spliced transcript expression predicted at day 3 was also highly correlated with the actual expression at day 7 (Pearson correlation coefficients, 0.86–0.99; mean, 0.94 between subject samples; Supplementary Fig. 2a) indicating that predictions can be valid over a timespan of multiple days. When we repeated this analysis for the SARS-CoV-2 dataset, there were also high

levels of correlation between these timepoints (mean: 0.96, range: 0.82–0.99, Supplementary Fig. 2b).

The genes identified by the maSigPro analysis were then used to generate fate maps with the RNA velocity of each subject at each timepoint indicating their predicted future transcriptomic state (Fig. 2g, h). Arrows on these fate maps represent the magnitude and direction of RNA velocity for each sample across all relevant genes in the depicted transcriptomic space. The RNA velocities are highly

**Fig. 1 | The blood transcriptome response to respiratory viral challenge.** Analysis of blood RNA in subjects inoculated with influenza A virus (**a**, **b**, **e**, **g**, **i**, **k**, **m**) or SARS-CoV-2 (**c**, **d**, **f**, **h**, **j**, **l**, **n**). **a**, **c** Schematic of blood draws. **b**, **d** Characteristics of participants. **e**, **f** Principal component analysis (PCA) of the spliced transcriptome across time following influenza ($n = 138$ samples, 21,647 genes, **e** and SARS-CoV-2 inoculation ($n = 257$ samples, 23,903 genes, **f**). **g** Volcano plots showing differentially expressed genes (DEGs) in day 2 samples between subjects first testing positive for influenza at day 1 ($n = 6$) vs those persistently negative ($n = 6$; 2988 DEGs, top), and in day 3 samples for subjects first testing positive for influenza on day 2 ($n = 9$) vs those persistently negative ($n = 6$; 892 DEGs, bottom). **h** Volcano plot showing DEGs in day 3 samples from subjects who were positive for SARS-CoV-2 ($n = 17$) vs those persistently negative ($n = 9$; 406 DEGs). Benjamini-Hochberg adjusted $p$ values were calculated after two-sided gene-wise linear model likelihood ratio tests (**g**, **h**). Dots represent individual genes; red, statistically significant DEGs with absolute $\log_2$ fold change (FC) > 1; green, non-significant genes with absolute $\log_2$FC > 1;

blue, statistically significant DEGs with absolute $\log_2$FC ≤ 1; grey, non-significant genes with absolute $\log_2$FC ≤ 1. **i**, **j** PCA plots of the unique multi-exon DEGs identified in day 2 and 3 samples of influenza challenge ($n = 46$ samples, 2115 genes, **i**) and day 3 of SARS-CoV-2 challenge ($n = 26$ samples, 294 genes, **j**). **k**, **l** Spliced transcript expression of two genes with greatest changes over time between infected subjects (red triangles) and PCR-negative subjects (blue circles; influenza, $n = 138$ samples; SARS-CoV-2, $n = 257$ samples). Shading represents 95% confidence intervals around the locally estimated scatterplot smoothing regression lines. **m**, **n** PCA plots using all genes with significant changes in spliced transcript expression over time between infected and uninfected subjects (**m**, influenza, $n = 138$ samples, 1563-genes; **n** SARS-CoV-2, $n = 257$ samples, 2700 genes). Red arrow illustrates temporal ordering of samples, blue circle identifies samples from subjects who remained PCR-negative throughout. Day 0 (pre) and (post) indicate samples pre- and post-inoculation, respectively. Source data are provided as a Source Data file.

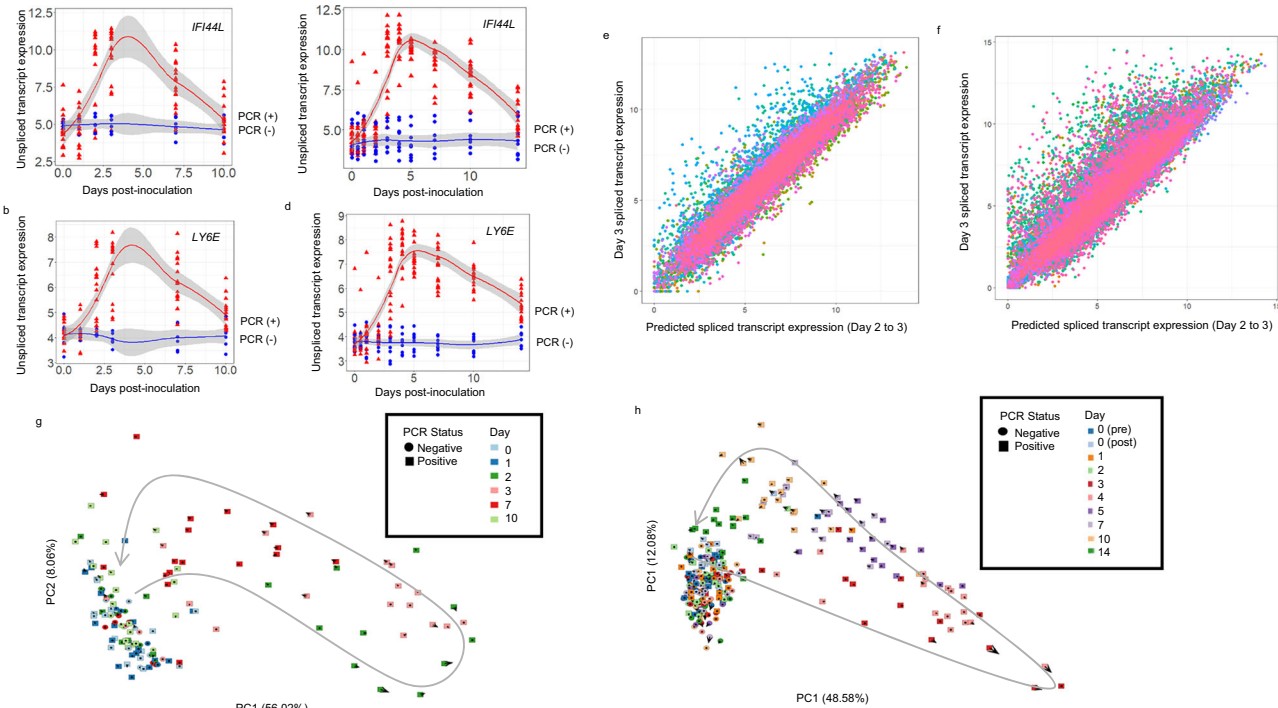

**Fig. 2 | Changes in the balance of unspliced and spliced transcripts in blood are predictive of future transcriptomic and disease states.** **a**–**d** Unspliced transcript expression of *IFI44L* (**a**, **c**) and *LY6E* (**b**, **d**) over time in influenza A (**a**, **b**; $n = 138$ samples) and SARS-CoV-2 (**c**, **d**; $n = 257$ samples) infections, respectively. Shading represents 95% confidence intervals around the locally estimated scatterplot smoothing regression lines. Red triangles represent PCR-positive (infected) participant samples. Blue circles represent PCR-negative participant samples. **e**, **f** predicted future spliced transcript expression (calculated at day 2) versus actual

spliced transcript expression at day 3 for influenza (1563-genes, **e**) and SARS-CoV-2 (2700 genes, **f**). Genes (dots) were selected using maSigPro and are coloured by sample. **g**, **h** RNA velocity fate maps for influenza ($n = 138$ samples, 1563-genes, **g**) and SARS-CoV-2 ($n = 257$ samples, 2700 genes, **h**) studies. RNA velocity arrows indicate predicted transcript trajectories. Transition probability number of neighbour values: 45 (**g**) and 25 (**h**). Grey arrow indicates temporal progression of samples. PC, principal component. Source data are provided as a Source Data file.

concordant with the observed temporal progression of the transcriptome along diverging trajectories for infected and uninfected subjects.

Together, these analyses illustrate that RNA velocities calculated with VeloCD predict both future transcriptomic and disease states.

## Outcome prediction in controlled human infection

To move towards clinical application, we studied how VeloCD can provide quantitative predictions of transition to defined disease states. To do this, VeloCD uses reference transcriptome data from subjects who have the disease states of interest and then determines the probability of a test subject transitioning to each group based on their RNA velocity values (Fig. 3a, b). We refer to these values as Transition Probabilities (TPs), which represent sample to group probabilities

calculated from RNA velocity values. We assessed how well VeloCD predicts future infection status in the CHIM subjects using only samples collected at day 1 post-inoculation.

For the influenza study, samples taken the day after the first positive PCR result were used for the infected reference group ($n = 15$) and the day 3 samples from the subjects who remained PCR-negative throughout were used as the uninfected reference group ($n = 6$). The day 3 sample from the single infected subject who first tested positive on day 3 was also included in the infected reference ($n = 16$ in total, Fig. 3a). The single subject who became PCR-positive on day 4 was excluded from the reference groups because no blood samples were collected on days 4 or 5. For the SARS-CoV-2 study, the day 3 samples from PCR-positive individuals were used as the infected reference group ($n = 17$) and the day 3 samples from those remaining

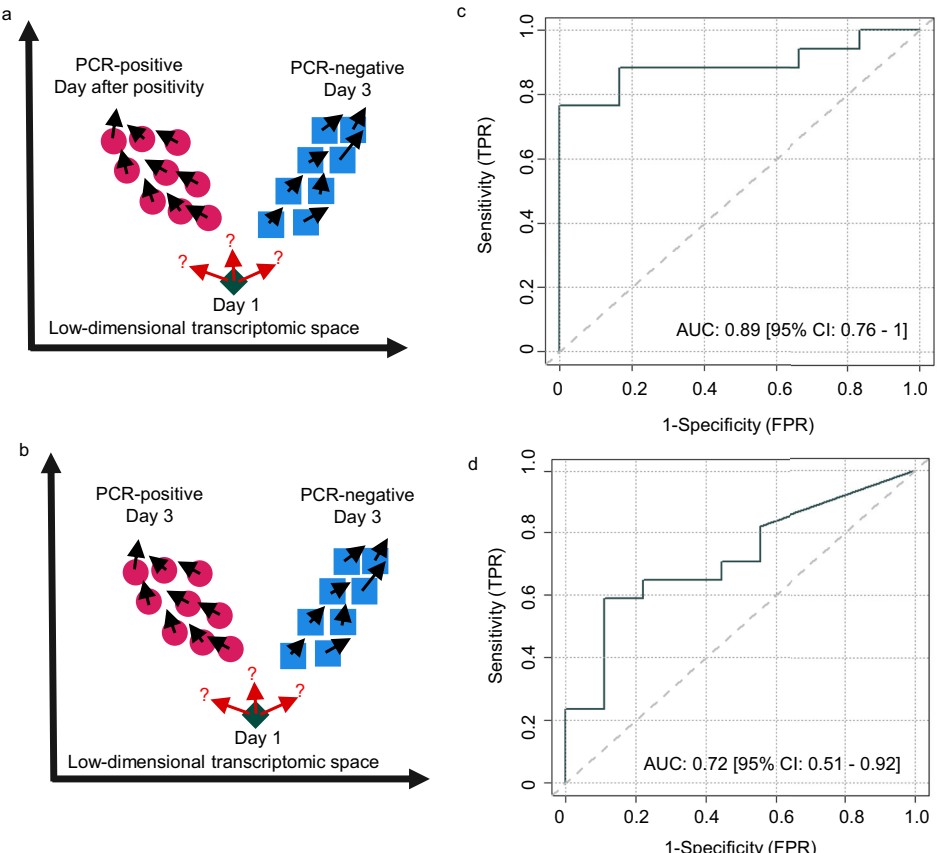

**Fig. 3 | The prognostic potential of VeloCD in influenza A and SARS-CoV-2 controlled human infections. a**, **b** Schematics illustrating the drop-one-in approach, where one test sample (green diamond) is dropped into VeloCD alongside samples from the reference groups (red circles, blue squares). **c** Performance of the 3-gene influenza signature to predict future PCR-status from the day 1 post-inoculation sample in the influenza study (23 participants, generated using a transition probability number of neighbours value of 11 and principal component analysis-based embeddings). **d** Performance of the 232-gene signature to predict future PCR status from day 1 post-inoculation in the SARS-CoV-2 cohort (26 participants, generated using a TPNN of 6 and PCA-based fate maps). AUC, area under the curve; CI, confidence interval. Source data are provided as a Source Data file.

PCR-negative throughout as the uninfected reference group ($n = 9$, Fig. 3b).

Many genes expressed in blood do not vary in expression with changes in disease status, whilst other genes may have non-monotonic expression dynamics which are poorly modelled by VeloCD[30]. Inclusion of these genes in RNA velocity calculations may mask the signal of more informative genes which are changing in response to infection. Consistent with this, we have found that VeloCD works best if it uses RNA transcripts which have been selected to distinguish between reference disease groups, and which vary in expression in association with diverging disease trajectories (Supplementary Fig. 3).

Therefore, we developed a more rigorous framework to identify genes with differential expression between the reference groups and with concordance between their RNA velocity (at day $n$) and measured change in spliced transcript expression (from days $n$ to $n + 1$). For the influenza study, we then used least absolute shrinkage and selection operator (LASSO) regression to identify a smaller group of highly informative genes: BCL2 Antagonist/Killer 1 (*BAK1*), Apolipoprotein L3 (*APOL3*), SLC9A3 Antisense RNA 1 (*SLC9A3-AS1*), Phospholipase A and Acyltransferase 4 (*PLAAT4*), H2A.Z Variant Histone 2 (*H2AZ2*), from which we picked the three genes (*BAK1*, *APOL3*, *SLC9A3-AS1*) contributing most to the first three Principal Components (PCs; Supplementary Fig. 4, Supplementary Table 2). *BAK1* encodes the proapoptotic BCL2 Antagonist/Killer 1 protein[31]. *APOL3* is a member of the apolipoprotein L gene family, encoding a protein involved in lipid movement and binding as well as interferon stimulated immunity[32].

*SLC9A3-AS1* is a long non-coding RNA gene[33]. Phase portraits of spliced and unspliced transcript expression further confirmed that *BAK1* and *APOL3* had desirable patterns of transcript expression (Supplementary Fig. 5).

*BAK1*, *APOL3* and *SLC9A3-AS1* were then input into VeloCD to calculate the probability of each subject becoming influenza PCR-positive, based on their day 1 blood sample. In each run, one test subject was "dropped-in" to VeloCD with the reference samples (Figs. 3a, b) with the removal of that subject's own sample from the reference prior to each run, to prevent bias due to sample donor identity. This process was repeated for each test subject ($n = 23$), across each possible number of neighbouring samples (the transition probability number of neighbours (TPNN) hyperparameter, see Methods). TPs were then calculated by VeloCD (see Methods) as the probability of each test subject transitioning to each outcome group (infected or PCR-negative). Sensitivity and specificity were then calculated by varying the probability threshold required to classify a subject as becoming infected (PCR-positive). Collectively, these steps enable the use of VeloCD to make quantitative predictions of future disease state.

A TPNN value of 11 was optimal to predict infection status up to 3 days into the future (Area Under the Receiver Operating Characteristic (AUROC) curve value of 0.89; 95% Confidence Interval: 0.76-1; Fig. 3c). The range of AUROC values across TPNN (range 2-19) hyperparameter runs was 0.62-0.89 (Supplementary Fig. 6a). We investigated if these probabilities correlated with future symptom scores for

the study participants. Using day 1 samples, the VeloCD-predicted probabilities of becoming PCR positive (by the end of the study), were moderately correlated with symptom score at day 2 ($p$ value = 0.023, Pearson correlation coefficient: 0.47, $n$ = 23 total, $n$ = 14 had non-zero symptom scores, Supplementary Fig. 7).

The process of differential expression and discordance-concordance (DISCO) analysis was repeated for the SARS-CoV-2 dataset (Supplementary Fig. 8), identifying 232-genes (Supplementary Data 1). The functions of these genes were assessed using Gene Ontology (GO) analysis, identifying 241 significant terms (after multiple testing correction) including defence response to viruses and response to other organism (Supplementary Data 2).

Neither LASSO regression nor selection by contributions of each gene in PCA yielded a smaller set of highly discriminatory genes, therefore all 232 genes were used in VeloCD to predict future SARS-CoV-2 PCR status. At the optimal TPNN of 6, the AUROC was 0.72 (95% CI: 0.51-0.92; Fig. 3d), predicting infection status moderately well up to 48-hours into the future. The range of AUROC values between different TPNN (range 2-25) hyperparameter runs was 0.53-0.72 (Supplementary Fig. 6b). Interestingly, applying this signature to the day 0 (post-inoculation) samples, yielded a predictive performance similar to when day 1 samples were used as the test set (optimal AUROC, 0.73, 95% CI: 0.51-0.95, TPNN: 10, full range of AUROCs: 0.52-0.73), indicating potential to predict up to 3 days into the future.

Since influenza and SARS-CoV-2 are both viruses with single stranded RNA genomes[34,35], which activate many common immune pathways[36], we tested the predictor genes identified in each dataset on the other. The 3-genes selected to predict influenza infection performed moderately well in the SARS-CoV-2 dataset, with an AUROC of 0.76 (95% CI: 0.57-0.96, TPNN: 16, Supplementary Fig. 9; AUROC range 0.56-0.76 across TPPNs 2 to 25), demonstrating validation in an independent dataset. Two of the 3 genes showed clear patterns of expression change across time in both datasets with spliced transcript expression changes in the SARS-CoV-2 infected subjects lagging slightly behind those in the influenza infected subjects (Supplementary Fig. 10), consistent with the divergence between the infected and PCR-negative blood transcriptomes appearing to occur from day 3 and day 2, respectively (Fig. 1e, f). Of the 232 genes selected to predict SARS-CoV-2 infection, 216 were expressed in the influenza dataset. These did not perform much better than chance to predict future infection status from day 1 samples in the influenza dataset (TPNN: 5, AUROC: 0.53, 95% Confidence Interval: 0.11-0.84), possibly because they were selected to identify the slightly later onset of the transcriptomic response to SARS-CoV-2 infection. If day 2 samples from the influenza study were used as the test set, AUROC values were between 0.75-0.88 (optimal performance: 0.88, 95% CI: 0.72-1, TPNN: 4) across hyperparameters.

Taken together, our results in CHIMs demonstrate that it is possible to make quantitative predictions of future disease state based on RNA velocity of blood. A potential application could be early identification and isolation of individuals who have been infected following exposure to high consequence pathogens, before the pathogen itself becomes detectable or transmissible to others.

### Generalizability of RNA velocity to natural SARS-CoV-2 infection

The standardized data from the CHIM studies was essential for validation of the principles underpinning VeloCD, but it is important to assess whether the same principles can be generalized to naturally occurring infection. Therefore, we used longitudinal RNA-Seq data from 56 participants (Supplementary Fig. 11a) in the household contact Integrated Network for Surveillance, Trials and Investigations into COVID-19 Transmission (INSTINCT) study[37], to assess expression dynamics of genes identified in the SARS-CoV-2 CHIM. In the INSTINCT dataset, all SARS-CoV-2 infected study participants were already PCR positive at the time of the first sample, so we could not test predictive

performance of the 232-gene signature, discovered in the SARS-CoV-2 CHIM dataset. Instead, we examined how well the RNA velocity fate map captures the resolution of infection of PCR positive participants, with reference to the transcriptomic state of 32 uninfected participants.

Focussing initially on *IFI44L* and *Ly6E*, we identified patterns of spliced and unspliced transcript expression (Supplementary Fig. 11b and c) similar to those in the later stages of infection in the SARS-CoV-2 CHIM study (Fig. 1k-l). We used the 232 genes in the predictive signature from the SARS-CoV-2 CHIM-analysis, to examine disease trajectories in a fate map constructed from all timepoint samples available in the INSTINCT dataset (Supplementary Fig. 11d). The first samples from the infected subjects were well separated from the uninfected subjects, consistent with the infected subjects already being PCR positive at the time of first sampling. There was a general trend for the first time point samples from infected individuals to have velocities away from the uninfected subjects, before subsequent sample positions and velocities returning towards the uninfected groups, consistent with the dynamics of *IFI44L* and *Ly6E*. These findings demonstrate that there is generalisability of trajectories indicated by RNA velocity between CHIM and naturally acquired infection.

### RNA velocity predicts onset of TB-IRIS in a clinical trial

Whilst the serial sampling in the CHIM and INSTINCT studies enable validation of the assumptions underpinning VeloCD, longitudinal samples should not be a requirement for VeloCD to make informative predictions. Therefore, we next investigated whether VeloCD could be used to predict the onset of a different disease, TB-IRIS, using samples from a single time-point.

TB-IRIS is a paradoxical immunopathological reaction that develops in ~18% of patients with HIV-associated TB shortly after initiation of anti-retroviral treatment (ART)[38]. It is characterized by systemic hyperinflammation and new or recurrent symptoms at sites of TB disease[39]. Recent studies have demonstrated that TB-IRIS has a distinct host blood transcriptomic signature with some prognostic potential[40]. We analysed whole-blood transcriptomes of subjects ($n$ = 95) from the placebo arm of a clinical trial[41], in which subjects with HIV-associated TB who had received TB treatment for less than 30 days were started on ART and randomized to receive either prednisone or placebo[41] (Supplementary Table 1).

Using samples collected two weeks after ART initiation, which is the peak time of onset of TB-IRIS, we assembled three groups–a test group who developed TB-IRIS after the two week blood sample (range 15–37 days after ART initiation, $n$ = 8), a reference group who had already developed TB-IRIS within the first 14 days of the trial ($n$ = 36), and a reference group who did not develop TB-IRIS during at least 12 weeks of follow-up (no-TB-IRIS, $n$ = 51). The test group was used to examine whether RNA velocity could indicate the future occurrence of TB-IRIS in participants who had not yet developed the syndrome at the two-week sampling time-point.

PCA using spliced transcript expression of all genes demonstrated modest segregation of the TB-IRIS and no TB-IRIS patients (Fig. 4a), with those yet to develop TB-IRIS located between the two reference groups. DEA identified 5,108 DEGs between the TB-IRIS and no-TB-IRIS groups (Fig. 4b), and PCA using these DEGs produced somewhat clearer segregation based on TB-IRIS status (Fig. 4c). We then applied the DEA and DISCO analysis steps of the framework described above to this dataset (Supplementary Fig. 12), identifying 1,980 genes to use for prediction of TB-IRIS status. We briefly examined the biological functionality of these genes and identified 295 significant GO terms (after multiple testing correction) including regulation of biological process and protein binding (Supplementary Data 2).

Each test subject was combined separately with the reference subjects to predict their future status using VeloCD (Fig. 4d). The TPs of each test subject to the TB-IRIS group were collated across runs with

equivalent hyperparameter values (TPNN: 5-50). Seven of eight (87.5%) test subjects were predicted to develop TB-IRIS (TP threshold: 30%, TPNN: 30, PCA used to construct the fate maps). Across the full range of TPNN values (5-50), using the same 0.3 probability threshold, TB-IRIS was correctly predicted in a median of 75% of test subjects (Supplementary Fig. 13).

To illustrate the relationships between subjects and disease states and demonstrate the potential for VeloCD to utilize different low-dimensional embedding methods, a 3-dimensional uniform manifold approximation and projection (UMAP)-based RNA velocity fate map was generated using all test and reference subjects in the TB-IRIS dataset (Fig. 4e). This shows the two reference groups progressively cluster towards the ends of a continuum with velocity arrows predominantly pointing away from each other, whilst test subjects lie towards the middle with velocity arrows predominantly pointing towards the TB-IRIS group.

To formally assess how well RNA velocity predicts the onset of TB-IRIS in this dataset, we created a test set from 25 randomly selected subjects who did not develop TB-IRIS and the eight subjects who developed TB-IRIS after day 14, and re-ran analysis, selecting a new 56-gene signature (Supplementary Fig. 14a, Supplementary Data 1). This signature had an AUROC of 0.72 [95% CI: 0.47-0.96] (Supplementary Fig. 14b, TPNN: 18, Number of Neighbours (NN): 25, t-distributed stochastic neighbour embedding (tSNE)-based embedding). The full range of AUROCs across all TPNNs is shown in Supplementary Fig. 14c. GO term analysis of the 56 genes identified 256 terms significant before multiple testing correction (none after correction), with the top terms being "axonal fasciculation" and "lymphocyte chemotaxis across high endothelial venule" (Supplementary Data 2).

Despite the small and unbalanced size of the test groups and the long interval between sampling and the onset of TB-IRIS in many subjects, these results demonstrate the potential of using RNA velocity for prediction and exploration of disease trajectories, based on measurements at a single time-point.

## RNA velocity predicts response after treatment for IBD

A potential application of VeloCD could be early identification of whether an administered medication will cause a desired effect. We assessed this in a publicly available dataset of 13 IBD patients undergoing their first treatment course of anti-TNF treatment (infliximab infusions at weeks 0, 2, 6 and 14)[42]. Six of these subjects reached remission by week 14 post-treatment (Supplementary Fig. 15a). We investigated whether VeloCD could predict future remission status using the trajectory of the response 2 weeks after the first infliximab infusion. The reference sample set was the week 6 samples ($n = 12$), which showed 731 DEGs, between subjects achieving remission by week 14 ($n = 5$, with this timepoint sample available) and those not achieving remission ($n = 7$) (Supplementary Fig. 15b, c).

PCA performed using these DEGs showed clear separation of samples at week 6 by their final remission status (Supplementary Fig. 15d). DISCO analysis identified 18 of the 731 genes which across all 24 samples showed concordance between RNA velocity values and unspliced minus spliced transcript values (Supplementary Fig. 15e, Supplementary Table 3). Overall, there were 237 significant GO terms (before multiple testing correction; none were significant after correction) associated with these 18 genes, including purine-containing compound catabolic process and nucleoside phosphate catabolic process (Supplementary Data 2).

We then assessed prediction of remission status in a similar way to the preceding dataset analyses, iteratively combining each week 2 sample with the week 6 reference samples ($n = 11$, paired week 6 sample removed) (Supplementary Fig. 15f). Despite the small dataset, using VeloCD on the week 2 samples had reasonably good predictive performance for remission by week 14 with an AUROC of 0.83 [95% CI: 0.56-1] (Supplementary Fig. 15g-h). These findings suggest that RNA velocity could have application beyond acute infectious diseases and might be used to make early assessments of the effectiveness of treatments for chronic non-infectious diseases, before clinical evidence their impact becomes apparent.

## RNA velocity predicts severity of acute febrile illness

We have shown that VeloCD has potential as a prognostic tool when the timing of sampling relative to inoculation of a pathogen or initiation of treatment is controlled, but in real clinical environments, unwell individuals seek health care at different times and stages of illness. Many clinical decisions, such as admission to hospital, investigations, and medication choices, depend on perception of the likelihood that a patient's condition will improve or deteriorate. This is particularly the case for febrile children, who are often admitted to hospital because they appear unwell and there is concern that they may deteriorate, even though the majority have self-limiting viral illnesses and recover quickly. We therefore explored whether VeloCD could indicate trajectories of illness severity in febrile children.

For this analysis, we used whole-blood RNA-Seq data generated from 399 children with suspected infection, recruited to the Personalized Risk assessment in Febrile illness to Optimize Real-life Management across the European Union (PERFORM) study - a multi-country prospective observational study of children with suspected infection presenting to hospitals in Europe[43]. Children were recruited from Emergency Departments (EDs) and Paediatric Intensive Care Units (PICUs), resulting in a wide spectrum of severity ranging from mild (well enough to be sent home from ED), to moderate (admitted to hospital ward), and severe (admitted to PICU). A standardized phenotyping algorithm was applied to group children by proven or likely aetiology of illness[44].

The present analysis focused on subjects with proven or suspected bacterial ($n = 204$) or viral ($n = 195$) illness (probable, definitive, or "syndrome" categories of the diagnostic algorithm[44]; Table 1). Whilst accuracy of the aetiological diagnosis is expected to vary between these categories, definitive diagnosis is rarely known at the time of presentation and therefore should not be essential for predicting outcome. For each subject, whole-blood RNA-Seq data from the first sample collected at or shortly after the time of presentation was analysed (Supplementary Table 1). This dataset therefore captures the real-world clinical scenario of patients presenting to emergency departments and clinical decisions about hospital admission and treatment needing to be made, often before diagnosis is established.

PCA of spliced transcript expression of all multi-exon genes with detectable spliced and unspliced transcript expression from all subjects revealed some segregation by severity with the mild and severe groups most distinct from each other and the moderate severity group positioned between them, but there was considerable overlap (Fig. 5a). Unsurprisingly, across the entire cohort, the aetiology of infection (bacterial or viral) was related to disease severity, with the bacterial infections over-represented (71.4%) in the group recruited from PICU, compared to 51.1% and 33.3% of the moderate and mild illness groups, respectively (Table 1).

We created a test-set for later RNA velocity prediction, comprised of 7 subjects recruited in the ED but transferred promptly to the PICU (severe-illness, all bacterial), all 184 subjects admitted from ED to the ward (moderate illness), and 20 (10 bacterial, 10 viral) randomly selected mild-illness subjects. Using the remaining mild and severe subjects we sought to minimize the impact of aetiology on selection of genes for RNA velocity analysis, by performing separate DEAs between subjects with mild ($n = 68$) and severe ($n = 26$) viral infections, and between subjects with mild ($n = 29$) and severe ($n = 65$) bacterial infections, identifying 4915 and 7087 DEGs respectively (Fig. 5b). 2348 of these DEGs had concordant patterns of up- or down-regulation between the bacterial and viral groups and were taken forward. PCA

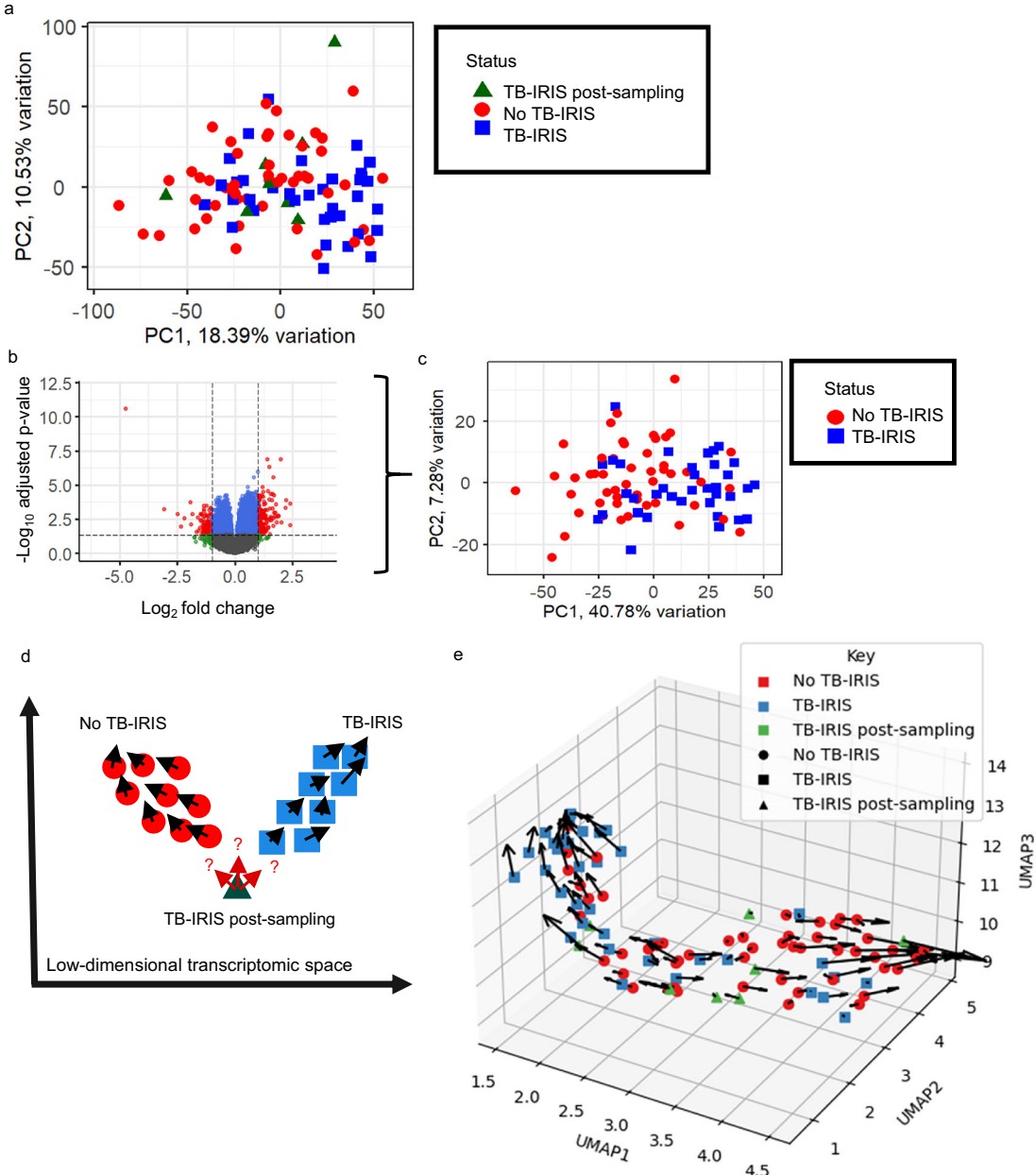

**Fig. 4 | RNA Velocity analysis of individuals at risk of tuberculosis-associated immune reconstitution inflammatory syndrome (TB-IRIS) following initiation of anti-retroviral therapy. a** Principal component analysis (PCA) plot of the spliced transcript expression of all multi-exon genes ($n = 95$, 23,041 genes; green triangles, participants who developed TB-IRIS after the two-week blood sample, $n = 8$; red circles, participants who never developed TB-IRIS, $n = 51$; blue squares, participants who developed TB-IRIS before the two-week blood sample, $n = 36$). **b** Volcano plot showing the differentially expressed genes (DEGs) for participants who developed TB-IRIS within two weeks of treatment versus those who never developed TB-IRIS (5,108 DEGs, red and blue dots above the horizontal line). Benjamini-Hochberg adjusted $p$ values were calculated after two-sided gene-wise

Wald statistics. Dots represent individual genes; red, statistically significant DEGs with absolute $\log_2$ fold change (FC) $> 1$; green, non-significant genes with absolute $\log_2 FC > 1$; blue, statistically significant DEGs with absolute $\log_2 FC \leq 1$; grey, non-significant genes with absolute $\log_2 FC \leq 1$. **c** PCA plot of the genes from **b** with spliced and unspliced transcript expression (3,448 genes). **d** Schematic illustrating the drop-one-in approach for the TB-IRIS dataset. **e** Uniform manifold approximation and projection (UMAP)-based RNA velocity fate map of all the samples in this cohort constructed using a 1,980-gene signature (transition probability number of neighbours, 48; number of neighbours, 25; UMAP with three-dimensions). RNA velocity arrows represent the future transcriptomic trajectories of each participant's sample point. PC principal component.

performed using this subset of 2348 genes with spliced and unspliced transcript expression revealed clearer segregation of mild and severe illness (Fig. 5c).

We then used a similar gene selection framework to the preceding analyses (Supplementary Fig. 16), incorporating LASSO regression to identify a smaller set of 59 genes (Supplementary Data 1) for quantitative prediction of future clinical state with VeloCD. We used these 59 genes to create a fate map, in which we also incorporated data from 19

healthy control subjects to illustrate their relationship to the mild and severe illness groups in transcriptomic space (Fig. 5d). This fate map illustrates diverging trajectories of severe and mild disease with the RNA velocity arrows of many mild subjects pointing towards the healthy controls, presumably indicating a trajectory towards recovery from illness. Interestingly, the mild bacterial illness subjects showed some segregation from the mild viral illness subjects, perhaps suggesting different sub-trajectories within mild illness.

**Table 1 | Characteristics of subjects from the Personalized Risk assessment in Febrile illness to Optimize Real-life Management (PERFORM) study (n = 399)**

| Diagnosis | Mild, n = 117 | | Moderate, n = 184 | | Severe, n = 98 | |
|---|---|---|---|---|---|---|
| | Bacterial n = 39 | Viral n = 78 | Bacterial n = 94 | Viral n = 90 | Bacterial n = 71 | Viral n = 27 |
| Sex, n (%) | | | | | | |
| Female | 14 (36%) | 44 (56%) | 42 (45%) | 39 (43%) | 25 (35%) | 8 (30%) |
| Male | 25 (64%) | 34 (44%) | 52 (55%) | 51 (57%) | 46 (65%) | 19 (70%) |
| Age in months, median (range) | 59 (4.7–205) | 76 (1.2–216) | 50 (0.13–196) | 36 (0.26-209) | 40 (0.59–207) | 14 (0.36–170.7) |
| Sampling Location, n (%) | | | | | | |
| Emergency Department | 39 (100%) | 78 (100%) | 94 (00%) | 90 (100%) | 6 (8.5%) | 1 (3.7%) |
| Paediatric Intensive Care Unit | 0 (0%) | 0 (0%) | 0 (0%) | 0 (0%) | 65 (92%) | 26 (96%) |
| Death, n (%) | | | | | | |
| Yes | 0 (0%) | 0 (0%) | 0 (0%) | 0 (0%) | 6 (8.5%) | 1 (3.7%) |
| No | 39 (100%) | 77 (99%) | 94 (100%) | 90 (100%) | 64 (90%) | 26 (96%) |
| Unknown | 0 (0%) | 1 (1.3%) | 0 (0%) | 0 (0%) | 1 (1.4%) | 0 (0%) |
| Ethnicity, n (%) | | | | | | |
| North/Mid/East European | 20 (51%) | 33 (42%) | 67 (71%) | 56 (62%) | 49 (69%) | 16 (59%) |
| South Asian (Indian, Pakistani, Bangladeshi, Tamil) | 1 (2.6%) | 4 (5.1%) | 1 (1.1%) | 4 (4.4%) | 0 (0%) | 3 (11%) |
| South European | 16 (41%) | 26 (33%) | 16 (17%) | 12 (13%) | 11 (15%) | 2 (7.4%) |
| Other | 1 (2.6%) | 11 (14%) | 9 (9.6%) | 14 (16%) | 9 (13%) | 5 (19%) |
| Unknown | 1 (2.6%) | 4 (5.1%) | 1 (1.1%) | 4 (4.4%) | 2 (2.8%) | 1 (3.7%) |
| Ill Appearance, n (%) | | | | | | |
| Yes | 8 (21%) | 14 (18%) | 41 (44%) | 38 (42%) | 55 (77%) | 20 (74%) |
| No | 25 (64%) | 60 (77%) | 43 (46%) | 42 (47%) | 5 (7.0%) | 3 (11%) |
| Unknown | 6 (15%) | 4 (5.1%) | 10 (11%) | 10 (11%) | 11 (15%) | 4 (15%) |
| Diagnosis category, n (%) | | | | | | |
| Definite **v**iral | 0 (0%) | 46 (59%) | 0 (0%) | 72 (80%) | 0 (0%) | 15 (56%) |
| Probable **v**iral | 0 (0%) | 27 (35%) | 0 (0%) | 9 (10%) | 0 (0%) | 1 (3.7%) |
| Viral **s**yndrome | 0 (0%) | 5 (6.4%) | 0 (0%) | 9 (10%) | 0 (0%) | 11 (41%) |
| Definite **b**acterial | 12 (31%) | 0 (0%) | 61 (65%) | 0 (0%) | 48 (68%) | 0 (0%) |
| Probable **b**acterial | 12 (31%) | 0 (0%) | 17 (18%) | 0 (0%) | 15 (21%) | 0 (0%) |
| Bacterial **s**yndrome | 15 (38%) | 0 (0%) | 16 (17%) | 0 (0%) | 8 (11%) | 0 (0%) |

Source data are provided as a Source Data file.

We briefly explored the biological functionality of these 59 genes but identified no significant GO terms after multiple testing correction. Before correction there were 427 significant GO terms including cyclooxygenase pathway, which has an important role in the inflammatory response (Supplementary Data 2).

Using the remaining mild illness cases ($n = 97$) and the PICU-recruited severe cases ($n = 91$) as reference groups, we then evaluated the predictive performance of VeloCD using the 59 gene signature (Fig. 5e). VeloCD was run for each test subject ($n = 211$). TPs to severe illness were calculated across independent runs for each subject using TPNN values from 10–70, and the median value was used for further analysis. The mild and severe subjects in this test set had very polarized TPs (Fig. 5f). Median TP to severe illness was 0.0035 (viral, 0.0018; bacterial, 0.023) in the mild illness group and 95% ($n = 19$) of these subjects had TPs below 0.25. The median TP values of the severe illness group (presenting to ED but requiring immediate transfer to PICU, $n = 7$) was 1.0 and 85.7% ($n = 6$) of these subjects had TPs above 0.75, suggesting that these probabilities could potentially be used as the basis of a risk stratification system (Fig. 5f). The complete range of TP values across TPNN runs is shown for the entire test cohort in Supplementary Fig. 17.

Next, we examined predictions in the subjects with moderate illness, who were admitted to the hospital ward. As might be expected, there was much more variability in TPs for these subjects suggesting more variable disease trajectories. Interestingly, the TPs of the moderate viral patients (median TP value, 0.0019) were much more skewed towards zero than their bacterial counterparts (median TP value: 0.043; two-sided Kolmogorov-Smirnov test $p$ value = 0.0028; Fig. 5f).

In this observational study, subjects received treatments determined by their attending physicians which would likely reduce the chances of those with moderate illness progressing to severe illness and limit meaningful categorical prediction. Consistent with this, only 3.8% ($n = 7$) of the moderate severity subjects required later admission to PICU (range: 1-9 days from ward admission), and predictive performance using VeloCD TPs was modest (AUROC 0.72, 95% CI: 0.51-0.92, Supplementary Fig. 18a). However, these 7 subjects had a median TP value of 0.36 (range: 0.0-1.0), which was significantly higher than the median TP value in those not requiring later admission to PICU ($n = 177$, median TP value: 0.0080, range 0.0-1.0, Wilcoxon signed rank $p$ value = 0.045).

We assessed whether moderate illness subjects with higher TPs had additional clinical evidence consistent with a more severe trajectory of illness, based on the other available information in this dataset. Baseline clinical features of subjects in a hypothetical higher-risk group (TP to severe disease $\geq 0.25$, $n = 43$) were not significantly different from those with lower TPs (TP < 0.25, $n = 141$) at time of presentation (Supplementary Table 4), although odds ratios above unity indicated a tendency for these subjects to have more features of severe illness, and have bacterial rather than viral infection. Mann-Whitney U tests identified significant differences in the distribution of baseline C-Reactive Protein (CRP; TP $\geq 0.25$, $n = 41$, median (IQR) 60.3 mg/L (16.6-166.7); TP < 0.25, $n = 131$, median (IQR) 13.2 mg/L (4.1-49), $p$ value = $1.67 \times 10^{-5}$)

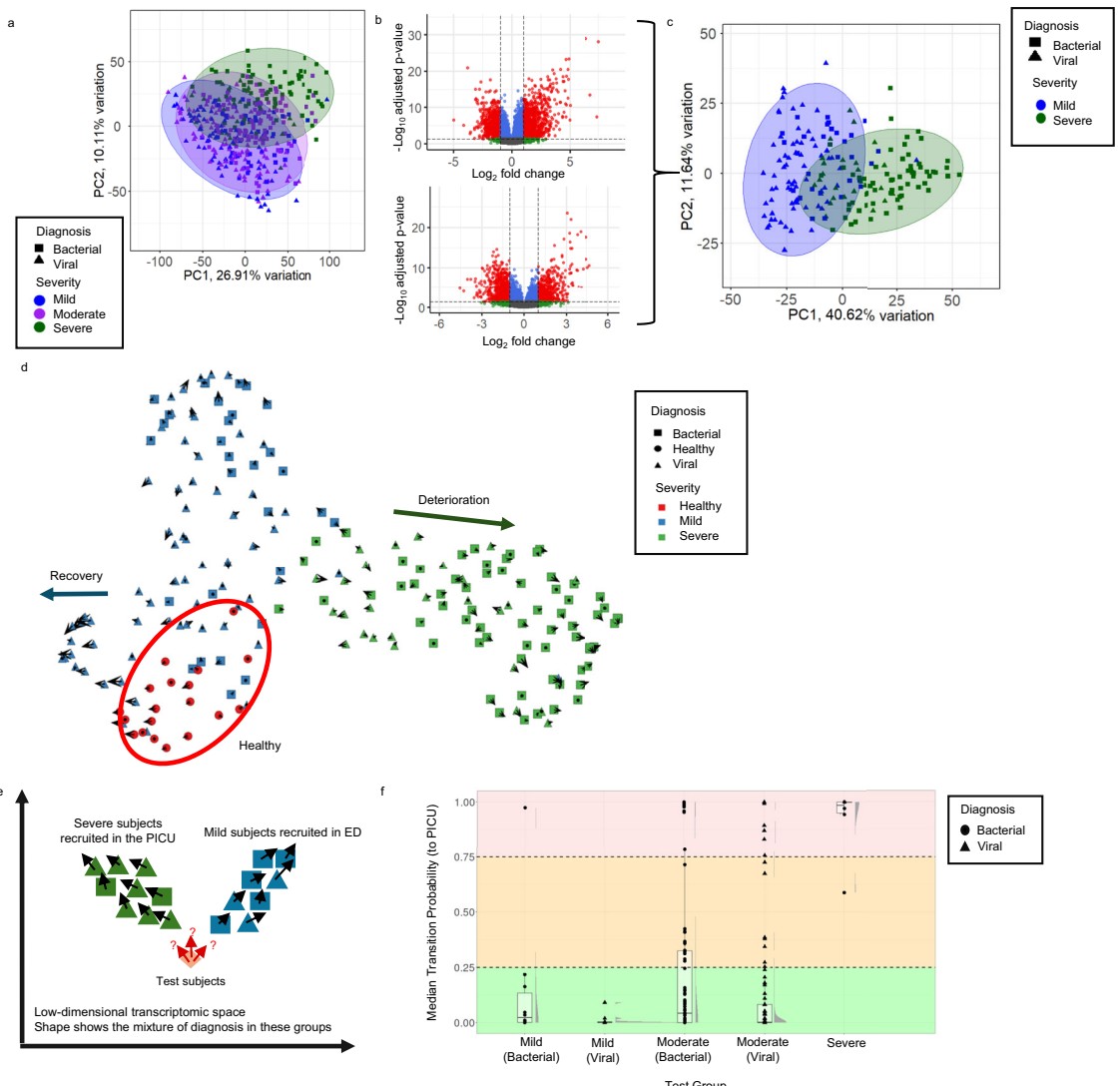

**Fig. 5 | RNA Velocity analysis of the Personalized Risk assessment in Febrile illness to Optimize Real-life Management (PERFORM) cohort. a** Principal component analysis (PCA) of spliced transcript expression of all genes (24,092) in PERFORM subjects (mild, $n = 117$; moderate, $n = 184$; severe, $n = 98$; bacterial, $n = 204$; viral, $n = 195$). Ellipses represent 95% confidence intervals around severity groups. **b** Volcano plots showing the differentially expressed genes (DEGs) between mild illness subjects recruited at the emergency department (ED) vs. severe illness subjects recruited in the paediatric intensive care unit (PICU) for bacterial (7087 DEGs, top) and viral illness (4915 DEGs, bottom). Benjamini-Hochberg adjusted $p$ values were calculated after two-sided gene-wise Wald statistics. Dots represent individual genes; red, statistically significant DEGs with absolute $\log_2$ fold change (FC) > 1; green, non-significant genes with absolute $\log_2$FC > 1; blue, statistically significant DEGs with absolute $\log_2$FC ≤ 1; grey, non-significant genes with absolute $\log_2$FC ≤ 1. **c** PCA plot of the spliced transcript expression of 1585 multi-exon DEGs with concordant $\log_2$FC values in the analyses shown in (**b**) for the mild illness subjects recruited at ED ($n = 97$) and the severe illness subjects recruited in PICU

($n = 91$). **d** UMAP-based RNA velocity fate map (using 59 genes) for 207 subjects including healthy controls ($n = 19$), mild illness subjects ($n = 97$), severe illness subjects recruited in the PICU ($n = 91$). Two alternative disease trajectories (deterioration and recovery) are illustrated. RNA velocity arrows represent future transcriptomic trajectories, generated with number of neighbours (NN) 25 and transition probability number of neighbours (TPNN) 80. **e** Schematic showing the drop-one-in approach applied to the PERFORM dataset. **f** Median transition probabilities (TPs, to the PICU) of 20 mild illness ($n = 10$ bacterial, $n = 10$ viral), 7 severe illness, and 184 moderate illness ($n = 90$ viral, $n = 94$ bacterial) subjects, calculated across TPNN values 10 to 70, projected into low-dimensional space using uniform manifold approximation and projection (UMAP; NN, 20). Background colour indicates hypothetical risk categories: high (red, TP ≥ 0.75), moderate (TP 0.25-0.75), and low (TP ≤ 0.25). The middle horizontal line of each boxplot represents the median; whiskers represent 1.5 times inter-quartile range. PC principal component. Source data are provided as a Source Data file.

and maximum CRP (TP ≥ 0.25, $n = 43$, median (IQR) 71.9 mg/L (25.1-196.7); TP < 0.25, $n = 137$, median (IQR) 21 mg/L (6-64); $p$ value = $4.81 \times 10^{-5}$; Supplementary Fig. 18b) between the TP groups. There was a significant CRP positive correlation between median TP values and the time from ward admission to hospital discharge (Spearman-rank $\rho = 0.17$, $n = 179$, $p$ value = 0.023) and the subjects with higher TP values were more likely to require surgical interventions (Odds Ratio = 4.02, $p$ value = 0.0052), all of which may be consistent with a more severe course of illness (Supplementary Table 4).

Next, we evaluated the performance of CRP (measured at the time of RNA sample collection) for distinguishing those who later transitioned to the PICU ($n = 5$, with this measurement available) versus those who did not ($n = 167$, with this measurement available). CRP had modest performance (AUROC 0.71, 95% CI 0.40-1.0) which was not significantly different to the performance of the VeloCD TPs (DeLong's, $p$ value = 0.97) (Supplementary Fig. 18a, c), but it should be noted that CRP is used to distinguish bacterial from viral illness in the PERFORM cohort, and all subjects transitioning from ward to PICU in the

test dataset had bacterial illness (with high maximum CRP throughout their illness).

We were interested to compare the predictions of future severity using VeloCD with risk stratification based on clinical data. We used the National Institute of Health and Care Excellence (NICE) criteria for stratification of risk of severe illness or death from sepsis[45] to classify children, based on clinical features, into three categories (red being high risk; amber, moderate to high risk; green, low risk; see methods). Although all 7 moderate group participants later requiring PICU admission were classified as red category, those who did not require PICU admission were 62.1% ($n = 110$) red, 11.3% ($n = 20$) amber, and 26.6% ($n = 47$) green—there was no significant difference in distribution of those requiring later PICU admission between categories (Fisher's exact test $p$ value = 0.19). However, there was a significant difference in the VeloCD TPs between the red category subjects ($n = 7$, median TP = 0.36) who did transition to PICU compared to the red category subjects who did not transition to the PICU ($n = 110$, median TP = 0.0048, Wilcoxon test $p$ value = 0.04). The NICE risk categories had moderate predictive performance for predicting transition to PICU (Supplementary Fig. 18d, AUROC: 0.69, 95% CI: 0.65-0.73). Together, this analysis indicates that RNA velocity is detecting differences between subjects which the NICE criteria do not distinguish.

We also used the PERFORM dataset to compare RNA velocity-based predictions with predictions from a static gene expression signature derived from the spliced transcript expression (but without considering RNA velocities). Taking the genes from the DEA of the reference set (PICU subjects vs. mild bacterial and mild viral infection subjects, separately) with concordant $log_2$ fold change (FC) values (2,348 genes), we performed LASSO-regression to reduce this to a 50-gene set that distinguished the mild from severe patients (Supplementary Data 1; $n = 97$ mild, $n = 91$ PICU-recruited). We then trained a generalized linear model on these genes and samples and derived scores for the test set subjects ($n = 20$ mild, $n = 7$ severe, $n = 184$ moderate). This signature had a numerically lower AUROC (0.68, 95% CI: 0.46−0.90, Supplementary Fig. 18e) than the RNA-velocity-based signature (Supplementary Fig. 18a). The scores were converted to probabilities (Supplementary Fig. 19) and compared between test groups. Unlike the RNA velocity-derived TPs, there was not a significant difference between the moderate illness subjects who later required PICU and those who did not ($p$ value = 0.09). The distribution of probabilities based on the static gene signature also appeared much more polarized relative to their RNA velocity-derived predictions, and there was no significant difference between the severe patients with bacterial or viral diagnoses ($p$ value = 1). Together, these findings indicate that RNA velocity likely augments prediction of future disease state in comparison to static analysis of spliced transcript expression alone.

Finally, we took advantage of this relatively large data set to examine the effect of varying reference set size on stability of VeloCD TPs. We performed triplicate random selections of 5, 10, 15, 20 or 25 samples for each of the 4 groups contributing to the reference set (total reference set sizes $n = 20, 40, 60, 80$ and $100$), and performed drop-one-in analysis for each test subject, at three NN values for each replicate dataset. The median TPs calculated for each NN across TPNN values were then subtracted from the equivalent median TPs generated using the full set of reference samples ($n = 188$). Overall, larger reference set sizes yielded more stable TP estimates (Supplementary Fig. 20), and these were most stable in the mild and severe illness test subjects, with more variation in the moderate severity group. Variation between replicates was noted, indicating that the composition of the reference set is also important.

Taken together with preceding results, these findings support the concept that RNA velocity can be used to make clinically meaningful predictions of future disease states, such as individuals most likely to recover or progress to severe illness, but further work may be needed to optimize reference datasets.

## Discussion

Predicting the trajectory of illness is a common challenge for clinicians. In acute illness, a cautious approach is often adopted to avoid missing patients who may develop life-threatening deterioration. This inevitably results in over-treatment and unnecessary admissions to hospital for many who would have milder illness trajectories, but also sometimes missed opportunities to intervene for those who appear well at presentation but subsequently deteriorate rapidly. More precise prediction of future disease state, severity, and response to treatment, could enable better risk stratification, expedited clinical management decisions, and better use of healthcare resources. We have shown that using RNA velocity to predict future clinical status may be a promising solution to this problem. We developed VeloCD as a tool to predict future disease states, over a timescale of days to weeks, from whole-blood bulk RNA-Seq data.

The concept of VeloCD is based on observations across multiple diseases, that the whole-blood bulk transcriptome reflects a patient's current disease state (both cause and severity) at the time of sampling, and that RNA velocities indicate future transcriptomic states, which therefore represent future disease states. In many cases, small sets of genes (signatures) have been identified to discriminate between different current disease states[5,11,25,46], which have led to the possibility of using these gene signatures in diagnostic tests[7,47]. Similarly, we have shown here that a gene selection framework (Supplementary Fig. 3) can be applied to identify smaller sets of genes which can be used for RNA velocity analysis to predict future disease states, across different diseases, study designs, and outcome measures. These examples illustrate potential clinical applications of RNA velocity, such as: early identification of individuals who are infected or uninfected after exposure to a pathogen (which could inform decisions about isolation or early treatment); identification of at risk individuals who will or won't develop a disease manifestation (determining the need for preventive treatment); early identification of response to treatment; and risk stratification based on trajectory of acute illness, which could help determine which patients can be sent home from the ED, should be admitted to hospital, or are highest risk and need closer monitoring or urgent intervention. Other applications could include improved understanding of pathogenesis and mechanisms of progression of acute illnesses, and stratified analyses of clinical trials to determine efficacy of treatments based on risk of outcome.

We expected that RNA velocity would only predict future disease state over a timescale of hours to a few days. It is interesting that the predictions of the onset of TB-IRIS and the response to infliximab in IBD, could be made days to weeks into the future, which may indicate additional utility if there is a lag between molecular changes in disease processes and their clinical manifestation.

Although we have used bulk RNA-Seq data as the basis for predictions in the current work, the identification of small sets of informative genes for calculation of RNA velocity opens the potential for translating this approach to other detection methods. Spliced and unspliced transcripts for each gene could be measured using quantitative PCR or loop mediated isothermal amplification methods, which would allow measurement on common and emerging diagnostic platforms[47]. Further studies will be required to assess such approaches.

Our study has several limitations. RNA velocity was originally developed for analyses of 1000 s of single cells rather than the tens or hundreds of samples available in our analyses. Whilst we divided our data in separate reference and test sets, we found evidence that the size and composition of the reference data set can impact predictions, and larger studies would allow more generalizable reference datasets, larger datasets for gene selection, and independent test and validation datasets for assessment of prediction accuracy. Although we showed

some stability of performance across a range of TPNN values, our prediction of future disease states was made using selected subsets of genes and hyperparameters, which risks over-optimistic performance. It will be important for future studies to evaluate prediction accuracy across independent datasets. In our analysis of the PERFORM study, relatively few outcome measures could be used to determine whether predictions of future severity were accurate for the moderately ill subjects who were admitted from ED to the ward – a group in which we believe there may be the most to gain from accurate prediction of future severity. Future observational studies will also face the challenge that association between prediction and observed outcome is modified by treatment given to patients and might need to collect more detailed data on early changes in physiological status and medical care to evaluate trajectories of illness.

This study has shown proof-of-concept that RNA velocity can be applied to whole blood samples to predict future disease states. The gene signatures selected in this study provide examples of potential utility but are not intended as definitive signatures for potential clinical use-cases. Further work will be required to develop and validate predictive tests based on this approach. If successful, this could lead to earlier detection and treatment of diseases and better risk stratification to aid clinical management decisions.

## Methods

### Ethical approval and conduct of human studies

The research was conducted in compliance with all relevant ethical regulations.

**Controlled human infection model (CHIM) studies.** Influenza and SARS-CoV-2 CHIM studies were performed as previously described[26,48], conducted in accordance with the protocol for each study, the Consensus ethical principles derived from international guidelines including the Declaration of Helsinki and Council for International Organizations of Medical Sciences (CIOMS) International Ethical Guidelines, applicable International Council for Harmonisation of Technical Requirements for Pharmaceuticals for Human Use (ICH) Good Clinical Practice guidelines, applicable laws and regulations. The studies were approved by UK Health Research Authority Research Ethics Committees (references: 11/LO/1836, 19/LO/1441, 20/UK/2001 and 20/UK/0002). For all studies, written informed consent was obtained.

In the influenza CHIM study, healthy adult volunteers aged 18-55 with microneutralising antibody titres of 1:20 or less for the inoculating virus were enroled and challenged with $5 \times 10^5$ tissue culture infectious dose 50 (TCID50) of influenza A/Belgium/4217/15 (H3N2; SGS, Antwerp, Belgium) by intranasal drops. Participants were quarantined for up to 10 days post-inoculation during which samples were taken, with outpatient follow-up continuing until 6 months post-inoculation. Generation of whole-blood paired-end RNA-Seq data is described in Rosenheim et al., 2023[49]. Data used in the present study originated from 23 individuals with 7 or 8 samples each taken at day 0 pre-inoculation and then day 1, 2, 3, 7, 10, 14 and 28 after inoculation with Influenza A/Belgium/4217/2015 (total $n = 178$ samples). Day 14 and 28 samples were not used in our analyses, leaving 138 samples from days 0-10.

Self-reporting symptom diaries were used to record symptoms throughout the study. On each day am and pm lower respiratory, upper respiratory and systemic systems scores were recorded on a severity scale between 0–3. These were then averaged (mean +/- standard deviation) for each day across 8 system types: sore throat, sneezing, cough, nasal discharge, nasal obstruction, headache, chillness and malaise using the Jackson system. From this a composite total score was calculated and used for analysis.

For the SARS-CoV-2 CHIM study, healthy adult volunteers aged 18–30 with no serological evidence of previous COVID-19 infection or vaccination were enroled. Participants were challenged with $10^2$ TCID50 of a D614G-containing pre-Alpha SARS-CoV-2 challenge virus by intranasal drops and quarantined for at least 14 days post-inoculation, with follow-up until 1-year post-inoculation. The generation of the CHIM SARS-CoV-2 whole-blood paired-end RNA-Seq data is described in Rosenheim et al., 2023[49]. Peripheral blood was collected from 35 participants on day 0 prior to inoculation and then on days 0, 1, 2, 3, 4, 5, 7, 10, 14 and 28 post-inoculation ($n = 375$ samples). The day 28 samples were not used in our analysis, and we also excluded all samples from two participants who sero-converted during pre-inoculation quarantine[27], and 7 subjects with equivocal infection status based on PCR results, leaving 257 samples from 26 individuals for analysis. Subjects classified as infected (PCR positive) were required to have multiple consecutive positive PCR tests and their first day of PCR-positivity was then defined by the first positive sample from nasal or throat swab. Subjects classified as uninfected (PCR negative) were required to be negative by PCR in throat and nose swabs throughout the entire course of the study.

**INSTINCT study.** The Integrated Network for Surveillance, Trials and Investigations into COVID-19 Transmission (INSTINCT) was a community-based observational study, approved by the UK Health Research Authority (ethics committee reference 20/NW/0231), in which primary cases and their household contacts were enroled soon after symptom onset in the primary case and followed longitudinally with collection of serial upper respiratory tract and blood samples. Details of the study design and generation of RNA-seq data have been published[37,49]. The RNA-seq dataset (EGAD50000000684) downloaded from the European Genome Phenome Archive consists of data from 56 individuals (Supplementary Fig. 11a) who were either PCR positive for SARS-CoV-2 at recruitment ($n = 24$), were persistently PCR-negative (uninfected) household contacts of someone who had tested positive ($n = 24$), or were non-contact uninfected controls ($n = 8$). There were no participants with RNA-seq data who were PCR-negative at recruitment and became PCR-positive during the study. RNA-Seq data was available from samples collected at days 0, 7, 8, 14 and 28 of follow-up ($n = 138$ samples).

**TB-IRIS study.** Transcriptomic analysis was conducted on samples from the PredART trial, a phase 3, randomized, double-blinded, placebo-controlled study to evaluate if prophylactic prednisone can safely prevent TB-IRIS occurrence in high-risk patients[41,50]. The study enroled 240 HIV-associated TB patients between August 2013 and February 2016 from 4 different TB clinics in Khayelitsha, South Africa. The study and protocol amendments were approved by the University of Cape Town Human Research Ethics Committee (HREC 136/2013). All participants provided written informed consent. Whole blood was collected in PAXgene tubes prior to commencing the randomized treatment, and at day 14 after starting treatment, from which total RNA was extracted using the PAXgene Blood RNA kit (PreAnalytiX). Total RNA sequencing library was prepared using Ovation Universal Human Blood kit (Tecan Genomics) and sequenced on an Illumina HiSeq4000 instrument using SR100 reactions with minimum 25 million reads. Only day 14 samples from the placebo group of the trial were used for the present study. One subject with sepsis and samples with less than 50% of their reads uniquely mapped to the human genome was excluded from further analysis, leaving 95 samples from 95 subjects.

**Inflammatory bowel disease treatment study.** Details of the original observational study design (approved by the ethics committee of the Christian-Albrechts-Universitat zu Kiel, A124/14 and AZ 156/03-2/13) and generation of RNA-seq data, have previously been published[42]. Raw RNA-Seq data from 13 inflammatory bowel disease (IBD) patients undergoing their first anti-TNF treatment course with infliximab infusions at weeks 0, 2, 6 and 14, were downloaded from GEO (GSE191328).

Six of these subjects reached remission by week 14 post-treatment. The data from blood samples collected at week 2 and week 6 were used for RNA velocity analysis.

**PERFORM.** The Personalized Risk assessment in Febrile illness to Optimise Real-life Management (PERFORM) study was a multi-centre, prospective, observational cohort study, seeking to improve the diagnosis of febrile illness in children across Europe (https://www.diamonds2020.eu/our-research-history/perform/).

Ethical approval was granted to the coordinating site (London – Central Research Ethics Committee: 16/LO/1684), and by local ethics committees for each recruitment site (Supplementary Table 5). Children (< 18 years old) presenting with fever, recent history of fever, or other features of suspected infection, who were considered ill enough by the medical team to warrant blood tests, were recruited from emergency departments, paediatric inpatient wards, and intensive care units, in 9 European countries, between August 2016 and December 2019. Full details of the study design and protocol have been published elsewhere[51,52]. Participants did not receive any compensation. All decisions on clinical investigation and management were made by attending clinicians in accordance with local practice. Each patient was assigned a final diagnostic classification by local study teams based on prospectively collected clinical and laboratory data and according to a standardized phenotyping algorithm[44,51]. Data entry and phenotyping were quality controlled by cross-site validation and automated checks within the study database[52]. For the present study, severity of illness was classified, based on level of care, as mild (well enough to be sent home from ED), moderate (admitted to hospital ward), or severe (admitted to PICU).

Blood was collected for research purposes at the same time as the first clinically indicated samples (wherever possible), and otherwise within 48 hours of admission. 2.5 ml whole blood was collected for RNA expression analysis in PAXgene tubes (for children below 1 year of age, 1 ml blood was collected into appropriately reduced volumes of PAXgene fluid) and frozen for later analysis. Total RNA was isolated using PAXgene miRNA blood extraction kits (Qiagen), and after additional DNAse treatment (Zymo Research), was sent for ribodepletion library preparation and RNA-Seq at The Wellcome Centre for Human Genetics in Oxford, United Kingdom, using a Novaseq6000 platform at 150 bp paired-end configuration, generating a raw read count of ~30 million reads per sample. Further details of the PERFORM RNA-Seq data generation have been published[53].

For the present analysis, we included subjects with RNA-Seq data whose samples were collected on presentation in the emergency department or collected in PICU (within 48 hours of presentation and >24 hours before PICU discharge), who had complete data on disposition (dates and locations of admission and discharge), and who were assigned one of the following diagnosis phenotypes: bacterial syndrome, probable bacterial, definite bacterial, viral syndrome, probable viral, definite viral. Two subjects each had two clinical episodes (>1 year apart), which were treated as separate events, giving a total of 399 infection episodes for analysis. We also included 19 non-infection control subjects recruited to PERFORM, predominantly from out-patient settings, who were well at the time of attendance and the reason for attendance was unlikely to affect the blood transcriptome, for example pre-surgical assessment for polydactyly or tonsillectomy.

The NICE criteria for stratification of risk of severe illness or death from sepsis were used to categorize risk of serious illness according to clinical features[45]. Children meeting the high-risk criteria (red category) were identified using the following variables: heart rate, respiratory rate, systolic blood pressure, temperature, capillary refill time, the presence of a non-blanching rash, an ill appearance to a healthcare professional and oxygen saturations, according to the guideline. The behaviour criteria, no response to social cues and does

not wake or if roused does not stay awake, were not specifically recorded in the PERFORM electronic case report form (eCRF). However, conscious level was documented via the Glasgow Coma Scale (GCS) and the Alert, Verbal, Pain, Unresponsive (AVPU) scale and therefore, any child, with a V, P or U on the AVPU scale or GCS less than or equal to 14, where this was a change from their normal conscious level, was assumed to satisfy the requirement in this category for being high-risk.

Where a variable was missing, the high-risk criteria was assumed not to be satisfied. In some instances, this may have led to children being categorized into a lower risk group. In particular, there was no field on the eCRF which captured a child who had mottled or ashen skin or who had cyanosis of the skin, tongue or lips. It is likely than many children with these findings on examination would have had other abnormalities that would have placed then in the high-risk category, but some children may have been miscategorized into a lower risk category because of the absence of this data. For all children, the data used for this categorisation was that recorded at triage.

## Sex and gender
Self-reported sex was reported in summary characteristics of each human study when available. We did not undertake any analyses stratified by sex because of insufficient statistical power.

## VeloCD
Spliced and unspliced transcript expression are input into VeloCD alongside two column metadata (maximum: 2, sample and group columns) files (CSV file format). VeloCD first constructs low-dimensional representations of the spliced transcript expression of the genes in a dataset using PCA[54], tSNE[55] or UMAP[56]. VeloCD then uses the spliced and unspliced transcript expression to calculate RNA velocity values, which indicate the direction and magnitude of future spliced transcript expression changes. These are then used to calculate transition probability (TP) values. UMAP and tSNE both use a Number of Neighbours (NN) hyperparameter (also called perplexity) to specify the balance between the conservation of the local and global structure of the data from high to low-dimensional transcriptomic space.

VeloCD builds on the methods established for the single-cell velocity tool, velocyto[15]. It works under the null hypothesis that no gene expression change is occurring (ie. gene expression is in a steady state). RNA velocities are then calculated as a deviation from this assumption: the modelled unspliced transcript expression subtracted from the actual unspliced transcript expression[15]. Positive velocity values indicate higher expression than modelled under the steady state assumption, indicating increasing gene expression, and negative values indicate decreasing gene expression. These values are then summed across genes to give overall RNA velocity values for each sample[15]. Velocity values are embedded as arrows on the low-dimensional transcriptomic space to give RNA velocity fate maps. A full description of these steps is given in La Manno et al., 2018[15]. Velocyto and VeloCD both perform these steps, but VeloCD has been further adapted for analysis of whole-blood RNA-Seq data. This adaptation process included the following: the removal of the spliced and unspliced normalisation steps from velocyto, removal of the grid-based velocity arrow visualisation, the introduction of UMAP-based fate maps, and the inclusion of functions that take the sample-to-sample TPs and calculate these across groups. All expression normalisation steps are now performed prior to the input of the spliced and unspliced transcript expression data into the algorithm (see RNA-Seq pre-processing).

To embed the RNA velocity values, VeloCD and velocyto both use RNA velocity values to calculate probabilities that each sample is transitioning to each other sample[15]. A nearest neighbour search identifies the *n* nearest neighbours of each sample (defined by the user

as the transition probability nearest neighbour (TPNN) value in VeloCD) in low-dimensional transcriptomic space. The probability values for each sample's non-neighbours are then converted to zero and the remaining non-zero values re-scaled to sum to 1. As additional functionality only present in VeloCD, this algorithm then takes the sample-to-sample TPs of each sample and sums them between (user-defined) groups, giving the sample-to-group TP values used to predict future disease outcomes of each subject.

For prediction analysis, VeloCD uses a drop-one-in approach where each sample in a user-defined test set is iteratively added into the spliced and unspliced expression files input into VeloCD. The algorithm is then run, and the TP values for the dropped in sample are then extracted in R and collated across TPNN runs. Low-dimensional embeddings are regenerated for each sample in a test set but remain the same across repeat analyses of the same sample using a range of TPNN values. For UMAP and tSNE the results shown were generated using a single number of neighbours (perplexity value).

VeloCD is implemented as a python-based downloadable software tool that can be run from the bash command line. Several R scripts are also called by the python code.

## Transcriptomics

**RNA-Seq pre-processing.** The fastq files from all RNA-Seq datasets were first quality control checked using FastQC[57], mapped using STAR[58] and quantified using featureCounts[59]. The strandedness of the data was confirmed using the infer_experiment.py programme of the RSeQC package[60]. Mapping and quantification statistics for all datasets are summarized in Supplementary Table 1. Any samples with a percentage of uniquely mapped reads ≤50% were excluded. When quality control revealed evidence of rRNA contamination, genes whose biotype was labelled as rRNA, Mt_rRNA, rRNA_pseudogene and ribozyme were removed from downstream analysis.

FeatureCounts was run using the meta feature option meaning reads were collated from all exons (per gene exon-level), introns (per gene intron-level) or features (gene) that overlapped a single gene into three separate gene-specific counts, using the human GRCh38 genome assembly fasta and GTF files from Ensembl. Prior to read mapping, the GTF file was edited to reduce the transcripts of the same gene down into the minimal non-overlapping set, with introns introduced between the new set of non-overlapping exons. This ensured a 1:1 ratio between the number of genes with one non-overlapping transcript and the number of transcript level features. This method also ensures that intronic sequences do not overlap any annotated exons of the same gene. When required, reads were summed for each sample across sequencing lanes.

The exon-level reads were used as an approximation for the spliced transcript expression[16,19]. The intron-level reads were used as an approximation for the unspliced transcript expression[16,19]. Genes with introns overlapping any exons of another gene on the same strand were removed from downstream analysis. Similarly, we attempted to remove genes with any introns labelled with the biotype retained intron based on the information available at the time of analysis. Only genes with both spliced and unspliced transcript expression were used for analysis. Spliced and unspliced transcript expression were then separately normalized for library size differences using the edgeR trimmed mean of m-values method and returned as $\log_2$ transformed counts per million values.

**Principal component analysis.** PCA was performed using the R packages: PCAtools[28], ggplot2[61], ggbiplot[62], factoMineR[63] and factoExtra[64]. The latter two packages were also used to extract the percentage contribution of each gene to each PC. The scale=FALSE argument was used for the generation of all PCA plots. We used PCA plots to exclude potential batch effects associated with known technical factors in each dataset.

**Differential expression analysis.** Genes with very low expression in each dataset were excluded from analysis by filtering on raw gene-level expression values to exclude genes which did not have counts of at least 5 reads in at least 3 samples. Differential expression analysis (DEA) of the CHIM datasets (Influenza and SARS-CoV-2) was performed using the glmFit and glmLRT functions from the edgeR package in R. The DESeq function from the DeSeq2 package was used to perform the DEA of the TB-IRIS and PERFORM datasets[65]. Methods were chosen based on their suitability to the datasets. PCA was used to determine which clinical variables were driving the axes of variation and should be included as covariates in their respective DEA. Sex was included as a covariate in the design matrices of the CHIM and TB-IRIS datasets. Age was also included in the design matrix of the SARS-CoV-2 CHIM dataset. X Inactive Specific Transcript (*XIST*) was later removed from the list of DEGs for DEAs of the TB-IRIS cohort. Gene-level expression values were required for this analysis because edgeR requires raw non-normalized expression[66]. All volcano plots were generated using the EnhancedVolcano package in R[67].

**Gene selection.** We developed a framework to select genes with desirable characteristics for RNA velocity-based prediction (Supplementary Fig. 3). First, DEA was used to identify genes with DE between the reference outcome groups of interest (eg., infected vs. PCR negative, step 1). Discordance-concordance (DISCO, step 2) analysis was then used to enhance identification of genes with concordance between calculated RNA velocity and changing expression, because it has previously been observed[68] that RNA velocity calculations can produce misleading results when there are very rapid changes in expression. This is because RNA velocity uses the expression of extreme samples (very low and very high expression) to model the steady state, which becomes inaccurate when there are a small number of samples with very high and/or low expression[68]. This was problematic when identifying genes that were suitable for VeloCD as many undergoing this change had nonsensical and ultimately useless RNA velocity values. By incorporating knowledge of future expression into this process the DISCO step identifies genes unaffected by this phenomenon. When serial measurements are unavailable, the unspliced minus spliced transcript expression can also be used as an approximation of the expected direction of RNA velocity. The gene selection framework then takes the intersection of the DEA and DISCO analysis as the initial gene set (step 3), which can then be reduced further using feature selection methods including LASSO regression[69] (glmnet package, optional step 4) and selecting those contributing most to the top PCs in PCA (optional, step 5).

**MaSigPro.** The R package maSigPro was used to perform backward selection to identify genes that were changing across time and also had divergent expression between groups. For the influenza dataset ($n = 138$), genes with low spliced and unspliced transcript expression were filtered prior to this analysis ($\geq 3$ in $\geq 3$ samples for both spliced and unspliced transcript expression). The participant IDs were included as the Replicate design matrix covariate in this analysis. For the SARS-CoV-2 CHIM dataset, the post-inoculation day 0 samples were assigned the value 0.5. Low expression pre-filtering was performed prior to this analysis (spliced and unspliced transcript expression $\geq 3$ in $\geq 3$ samples). The genes identified were then input into VeloCD for exploratory analysis (TPNN: 25).

**GO term analysis.** GO term enrichment analysis was performed using the goana() function (limma package) with the full set of genes with spliced transcript expression used as the background gene list for each analysis.

**Static prediction modelling.** For the static PERFORM gene signature, the glm() function was used to train the model and the predict()

function (response score type) was used derive new scores for the test set. Wilcoxon sign rank tests were used to compare the resulting probabilities between clinical groups.

## Statistical analysis

Fisher's exact tests (fisher.test() function in R) were used to determine the significance of the relationship between clinical features and the median TPs (to the PICU) grouped by a threshold value (≥0.25 or <0.25). A two-sided $p$ value cut-off of 0.05 was used to determine significance (base R stats package).

Pearson correlations were performed using the cor.test() function with the Pearson method argument in R. Similarly, Spearman-rank correlations were performed using this same function but with the spearman argument.

The wilcox.test() function was used to perform Mann-Whitney U tests. For all statistical tests, a two-sided $p$ value cut-off of 0.05 was used to determine significance. For the DEA, the Benjamini-Hochberg method was used to adjust for multiple testing and a threshold of 0.05 was used to determine statistical significance. Analysis generated from the maSigPro package returns Benjamini-Hochberg adjusted two-sided $p$ value. The ks.test() function was used to perform two-sided Kolmogorov-Smirnov tests between two distributions. Unless stated otherwise, all functions for statistical test were loaded from the base R stats package.

Sensitivity and specificity values were calculated across equivalent TPNN runs of VeloCD, and ROC curves, AUROC and 95% confidence intervals were constructed and calculated using the roc(), rocit(), auc() and ci.auc() functions from the ROCit[70] and pROC[71] R packages. The hyperparameter run with the highest AUROC was then selected as the best performance of the signature. The roc.test() function was used to perform DeLong's test to determine if two ROC curves generated using the roc() function were statistically significantly different.

## Visualisation

Raincloud plots were generated using ggthemes[72], ggdist[73,74] and ggplot2[61].

## Reporting summary

Further information on research design is available in the Nature Portfolio Reporting Summary linked to this article.

## Data availability

The RNA-Seq data from the TB-IRIS study have been deposited in the GEO database under accession number GSE274086. RNA-Seq data from the inflammatory bowel disease study have been deposited in the GEO database under accession number GSE191328. RNA-Seq data from the PERFORM subjects generated in this study have been deposited in the ArrayExpress database under accession code E-MTAB-14728, with the exception of 9 samples for which permission for public sharing of sequence data was not granted by participants. RNA-Seq data from the SARS-CoV-2 and influenza controlled human infection studies, and the INSTINCT study have been deposited in the European Genome-Phenome Archive (EGA) database under accession codes EGAD50000000942, EGAD50000000956, and EGAD50000000684, respectively. The data from these three studies are available under restricted access to comply with data privacy restrictions; access can be obtained by investigators whose proposed use is within the scope of the participant consent by submitting an application through EGA and subject to a data access agreement, as described previously[49]. Source data are provided with this paper.

## Code availability

VeloCD is available as a downloadable software tool (including documentation) in GitHub. It is accessible using the following link: https://github.com/DrClaireDunican/VeloCD/tree/main.

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

## Acknowledgements

We are grateful to the volunteers who participated in the studies described in this work. We gratefully acknowledge funding support from the following sources: the European Union's Horizon 2020 Research and Innovation programme (grant agreement numbers 668303 and 848196); the UK MRC (MR/R008922/1 to RPJL) and the UK Department for International Development (DFID) under the MRC/DFID Concordat agreement and is also part of the EDCTP2 programme supported by the European Union (MR/L006529/1 to A.J.C.); the UK research and Innovation Engineering and Physical Sciences Research Council (PhD and Postdoctoral prize fellowships to CD); the NIHR Imperial Biomedical Research Centre (BRC, support to CC CD and RJW); NIHR Biomedical Research Centre at University College London Hospitals (MN); the Wellcome Trust (207511/Z/17/Z to MN; 098316, 214321/Z/18/Z, and 203135/Z/16/Z to GM; 226817 to RJW); the Francis Crick Institute (which is supported by Wellcome (CC2112), MRC (CC2112) and Cancer Research UK (CC2112), support to RJW); the UK Vaccine Taskforce of the Department of Business, Energy and Industrial Strategy of Her Majesty's Government (BEIS); NIH Centres of Excellence for Influenza Research and Surveillance (CEIRS, contract number HHSN272201400008C); Defence Advanced Research Projects Agency (grant number 140D6319C00236); South African Research Chairs Initiative of the Department of Science and Technology and National Research Foundation (NRF) of South Africa (Grant No 64787, to GM); Taiwan's National Science and Technology Council (grant number 110-2923-B-006-001-MY4, to C-FS). We thank the following individuals and organizations for their invaluable contributions to development and implementation of the influenza and SARS-CoV-2 human challenge projects: Chris Woods (Duke University), SGS (Belgium) for supplying the influenza challenge agent, Andrew Catchpole (hVIVO), the NIHR Clinical Research Network staff at The Royal Bolton Hospital, Human Infection Challenge Network for Vaccine Development (HIC-Vac) and ISARIC4C Investigators (https://isaric4c.net/about/authors/). ISARIC4C is funded by the National Institute for Health Research (NIHR; award CO-CIN-01), the Medical Research Council (MRC; grant MC_PC_19059), the NIHR Health Protection Research Unit in Emerging and Zoonotic Infections at University of Liverpool in partnership with Public Health England (PHE), in collaboration with Liverpool School of Tropical Medicine and the University of Oxford (NIHR award 200907), Liverpool Experimental Cancer Medicine Centre provided infrastructure support for this research (grant reference C18616/A25153) and NIHR Health Protection Research Unit in Respiratory Infections (NIHR award 200927). The views expressed are those of the authors and not necessarily those of the NHS, the NIHR, DHSC, BEIS, the US Government, or Department of Defence.

## Author contributions

A.J.C., C.D., M.B., and M.K. conceived the study and interpreted analyses; D.H.-C., S. P., M.N., R.M., C.S., R.J.W., P.K.A.A., C.R.B., G.B., Uv.B., K.B.-P., E.D.C., L.J.M.C., G.D'.S., T.D., M.E., K.F., C.G.F., Mvd.F., I.G., L.K., M.K., T.K., F.M.-T., M.M.-T., S.N., S. Paulus, M.P., A.J.P., I.R.-C., A.R., L.J.S., N.A.S., C.-F.S., S.S., C.D.T., M.T., E.U., Fvd.V., C.L.V., M.V., S.Y., D.Z., W.Z., V.J.W., M.L., J.H., the PERFORM Consortium, R.P.J.L., G.M., and C.C. acquired data; C.D. and C.W. performed analyses; A.J.C. and C.D. draughted the manuscript; all authors reviewed the manuscript for important intellectual content and approved publication.

## Competing interests

An international patent application (126496PCT1) for a method to predict disease prognosis based on RNA velocity has been filed by Imperial College Innovations Limited, with A.J.C., C.D., M.B., and M.K. listed as inventors. The other authors have no relevant competing financial or non-financial interests as defined by Nature Portfolio.

## Additional information

Claire Dunican [1,2] ✉, Clare Wilson[1,2], Dominic Habgood-Coote [1], Suzanna Paterson[1], Mahdad Noursadeghi [3], Raymond Moseki[4], Cari Stek[4,5], Robert J. Wilkinson [1,4,6], Philipp K. A. Agyeman [7], Coco R. Beudeker[8], Giske Biesbroek[9], Ulrich von Both[10,11], Karen Brengel-Pesce[12,13], Enitan D. Carrol[14,15], Lachlan J. M. Coin [1,16], Giselle D'Souza[1], Tisham De [1], Marieke Emonts [17,18,19], Katy Fidler[20], Colin G. Fink [21], Michiel van der Flier [8,22], Ioanna Georgaki[23], Laura Kolberg [10], Mojca Kolnik[24], Taco Kuijpers[9,25], Federico Martinón-Torres [26,27], Marine Mommert-Tripon[12,13], Samuel Nichols [1], Stephane Paulus[28,29], Marko Pokorn[24,30,31], Andrew J. Pollard [28,29], Irene Rivero-Calle[26,27], Aleksandra Rudzate [32,33], Luregn J. Schlapbach[34,35], Nina A. Schweintzger [36], Ching-Fen Shen [37], Shrijana Shrestha [38], Chantal D. Tan[39], Maria Tsolia[23], Effua Usuf [40], Fabian van der Velden[17,18,19], Clementien L. Vermont[39,41], Marie Voice[21], Shunmay Yeung [42,43], Dace Zavadska [32,33], Werner Zenz[36], Victoria J. Wright [1,2], Michael Levin [1,2], Jethro Herberg[1,2], the PERFORM Consortium*, Rachel P. J. Lai[1], Graeme Meintjes[4,44,45], Christopher Chiu[1], Mauricio Barahona [46], Myrsini Kaforou [1,2] & Aubrey J. Cunnington [1,2] ✉

[1]Department of Infectious Disease, Imperial College London, London, UK. [2]Centre for Paediatrics and Child Health, Imperial College London, London, UK. [3]Division of Infection and Immunity, University College London, London, UK. [4]Wellcome Discovery Platform in Infection, Centre for Infectious Diseases Research in Africa and Institute of Infectious Disease and Molecular Medicine, University of Cape Town, Observatory, Cape Town, Republic of South Africa. [5]Department of Clinical Sciences, Institute of Tropical Medicine, Antwerp, Belgium. [6]Francis Crick Institute, London, UK. [7]Division of Pediatric Infectious Disease, Department of Pediatrics, Inselspital, Bern University Hospital, University of Bern, Bern, Switzerland. [8]Paediatric Infectious Diseases and Immunology, Wilhelmina Children's Hospital, University Medical Center Utrecht, Utrecht, the Netherlands. [9]Emma Children's Hospital, Amsterdam University Medical Center (Amsterdam UMC), location Academic Medical Center (AMC), Dept of Pediatric Immunology, Rheumatology and Infectious Diseases, University of Amsterdam, Amsterdam, the Netherlands. [10]Division Paediatric Infectious Diseases, Department of Pediatrics, Dr. von Hauner Children's Hospital, University Hospital, LMU Munich, Munich, Germany. [11]German Center for Infection Research (DZIF), Partner Site Munich, Munich, Germany. [12]Department of Recherche & Development, bioMérieux S.A., Marcy l'Etoile, Marcy-l'Étoile, France. [13]Joint research unit Hospice Civils de Lyon - bioMérieux, Centre Hospitalier Lyon Sud, Lyon, France. [14]Department of Clinical Infection, Microbiology and Immunology, University of Liverpool Institute of Infection, Veterinary and Ecological Sciences, Liverpool, UK. [15]Department of Infectious Diseases, Alder Hey Children's NHS Foundation Trust, Liverpool, UK. [16]Department of Microbiology and Immunology, The Peter Doherty Institute for Infection and Immunity, University of Melbourne, Melbourne, Australia. [17]Translational and Clinical Research Institute, Newcastle University, Newcastle upon Tyne, UK. [18]Great North Children's Hospital, Paediatric Immunology, Infectious Diseases & Allergy, Newcastle upon Tyne Hospitals NHS Foundation Trust, Newcastle upon Tyne, UK. [19]NIHR Newcastle Biomedical Research Centre based at Newcastle upon Tyne Hospitals NHS Trust and Newcastle University, Newcastle upon Tyne, UK. [20]Brighton and Sussex Medical School, University of Sussex, Sussex, UK. [21]Micropathology Ltd, Coventry, UK. [22]Amalia Children's Hospital, Radboud University Medical Center, Nijmegen, the Netherlands. [23]2nd Department of Pediatrics, National and Kapodistrian University of Athens, "P. and A. Kyriakou" Children's Hospital, Thivon and Levadias Goudi, Athens, Greece. [24]Department of Infectious Diseases, University Medical Centre Ljubljana, Ljubljana, Slovenia. [25]Sanquin Research Institute, & Landsteiner Laboratory at the AMC, University of Amsterdam, Amsterdam, the Netherlands. [26]Translational Pediatrics and Infectious Diseases, Pediatrics Department, Hospital Clínico Universitario de Santiago, Santiago de Compostela, Spain. [27]GENVIP Research Group, Instituto de Investigación Sanitaria de Santiago, Universidad de Santiago de Compostela, Galicia, Spain. [28]Oxford Vaccine Group, Department of Paediatrics, University of Oxford, Oxford, UK. [29]NIHR Oxford Biomedical Research Centre, Oxford, United Kingdom. [30]University Childrens' Hospital, University Medical Centre Ljubljana, Ljubljana, Slovenia. [31]Department of Infectious Diseases and Epidemiology, Faculty of Medicine, University of Ljubljana, Ljubljana, Slovenia. [32]Riga Stradins university, Riga, Latvia. [33]Children clinical university hospital, Riga, Latvia. [34]Department of Intensive Care and Neonatology, and Children's Research Center, University Children's Hospital Zurich, Zurich, Switzerland. [35]Child Health Research Centre, The University of Queensland, and Paediatric Intensive Care Unit, Queensland Children's Hospital, Brisbane, Australia. [36]Department of Pediatrics and Adolescent Medicine, Division of General Pediatrics, Medical University of Graz, Graz, Austria. [37]Department of Pediatrics, National Cheng Kung University Hospital, College of Medicine, National Cheng Kung University, Tainan, Taiwan. [38]Paediatric Research Unit, Patan Academy of Health Sciences, Kathmandu, Nepal. [39]Erasmus MC-Sophia Children's Hospital, Department of General Paediatrics, Rotterdam, the Netherlands. [40]Medical Research Council Unit The Gambia at LSHTM, Fajara, The Gambia. [41]Erasmus MC-Sophia Children's Hospital, Department of Paediatric Infectious Diseases & Immunology, Rotterdam, the Netherlands. [42]Faculty of Infectious and Tropical Disease, London School of Hygiene and Tropical Medicine, London, UK. [43]Faculty of Public Health and Policy, London School of Hygiene and Tropical Medicine, London, UK. [44]Department of Medicine, University of Cape Town, Cape Town, Republic of South Africa. [45]Blizard Institute, Faculty of Medicine and Dentistry, Queen Mary University of London, London, UK. [46]Department of Mathematics, Imperial College London, London, UK. *A list of authors and their affiliations appears at the end of the paper.
✉e-mail: claire.dunican14@imperial.ac.uk; a.cunnington@imperial.ac.uk

## the PERFORM Consortium

Michael Levin [1,2], Jethro Herberg[1,2], Myrsini Kaforou [1,2], Aubrey J. Cunnington [1,2]✉, Victoria J. Wright [1,2], Clare Wilson[1,2], Dominic Habgood-Coote [1], Giselle D'Souza[1], Tisham De [1], Samuel Nichols [1], Lucas Baumard[1], Evangelos Bellos[1], Giselle D'Souza[1], Rachel Galassini[1], Shea Hamilton[1,2], Clive Hoggart[1], Sara Hourmat[1], Heather Jackson[1], Naomi Lin[1], Ian Maconochie[1,2], Stephanie Menikou[1], Ruud Nijman[1,2], Ivonne Pena Paz[1], Oliver Powell[1], Priyen Shah[1], Ortensia Vito[1], Molly Stevens[47], Eunjung Kim[47], Nayoung Kim[47], Amina Abdulla[48], Ladan Ali[48], Sarah Darnell[48], Rikke Jorgensen[48], Sobia Mustafa[48], Salina Persand[48], Katy Fidler[20], Julia Dudley[49], Vivien Richmond[49], Emma Tagliavini[49], Enitan D. Carrol[14,15], Elizabeth Cocklin[50], Rebecca Jennings[51], Joanne Johnston[51], Aakash Khanijau[50], Simon Leigh[50], Nadia Lewis-Burke[50], Karen Newall[51], Sam Romaine[50], Andrew J. Pollard [28,29], Stephane Paulus[28,29], Rama Kandasamy[28,29], Michael J. Carter[28,29], Daniel O'Connor[28,29], Sagida Bibi[28,29], Dominic F. Kelly[28,29], Meeru Gurung[38],

Stephen Thorson[38], Imran Ansari[38], David R. Murdoch[52], Shrijana Shrestha [38], Zoe Oliver[53], Marieke Emonts [17,18,19], Emma Lim[18,19,54], Lucille Valentine[55], Karen Allen[56], Kathryn Bell[56], Adora Chan[56], Stephen Crulley[56], Kirsty Devine[56], Daniel Fabian[56], Sharon King[56], Paul McAlinden[56], Sam McDonald[56], Anne McDonnell[18,56], Ailsa Pickering[18,56], Evelyn Thomson[56], Amanda Wood[56], Diane Wallia[56], Phil Woodsford[56], Frances Baxter[56], Ashley Bell[56], Mathew Rhodes[56], Rachel Agbeko[57], Christine Mackerness[57], Bryan Baas[18], Lieke Kloosterhuis[18], Wilma Oosthoek[18], Tasnim Arif[58], Joshua Bennet[18], Kalvin Collings[18], Ilona van der Giessen[18], Alex Martin[18], Aqeela Rashid[58], Emily Rowlands[18], Gabriella de Vries[18], Fabian van der Velden[17,18,19], Joshua Soon[18], Mike Martin[59], Ravi Mistry[18], Shunmay Yeung [42,43], Juan Emmanuel Dewez[42], Martin Hibberd[42], David Bath[43], Alec Miners[44], Elizabeth Fitchett[42], Colin G. Fink [21], Marie Voice[21], Leo Calvo-Bado[21], Federico Martinón-Torres [26,27], Antonio Salas[26,27,60,61], Fernando Álvez González[26,27], Cristina Balo Farto[26,27], Ruth Barral-Arca[26,27,60,61], María Barreiro Castro[26,27], Xabier Bello[26,27,60,61], Mirian Ben García[26,27], Sandra Carnota[26,27], Miriam Cebey-López[26,27], María José Curras-Tuala[26,27,60,61], Carlos Durán Suárez[26,27], Luisa García Vicente[26,27], Alberto Gómez-Carballa[26,27,60,61], Jose Gómez Rial[26,27], Pilar Leboráns Iglesias[26,27], Nazareth Martinón-Torres[26,27], José María Martinón Sánchez[26,27], Belén Mosquera Pérez[26,27], Jacobo Pardo-Seco[26,27,60,61], Lidia Piñeiro Rodríguez[26,27], Sara Pischedda[26,27,60,61], Sara Rey Vázquez[26,27], Irene Rivero Calle[26,27], Carmen Rodríguez-Tenreiro[26,27], Lorenzo Redondo-Collazo[26,27], Miguel Sadiki Ora[26,27], Sonia Serén Fernández[26,27], Cristina Serén Trasorras[26,27], Marisol Vilas Iglesias[26,27], Chantal D. Tan[39], Clementien L. Vermont[39,41], Henriëtte A. Moll[39], Dorine M. Borensztajn[39], Nienke N. Hagedoorn[39], Joany Zachariasse[39], W. Dik[62], Michiel van der Flier[8,22], Ronald de Groot[63], Marien I. de Jonge[63], Koen van Aerde[22], Wynand Alkema[63], Bryan van den Broek[63], Jolein Gloerich[63], Alain J. van Gool[63], Stefanie Henriet[22], Martijn Huijnen[63], Ria Philipsen[63], Esther Willems[63], G. P. J. M. Gerrits[64], M. van Leur[64], J. Heidema[65], L. de Haan[22], C. J. Miedema[66], C. Neeleman[63], C. C. Obihara[67], G. A. Tramper-Stranders[67,68], Taco Kuijpers[9,25], Giske Biesbroek[9], Ilse Jongerius[25], J. M. van den Berg[9], D. Schonenberg[9], A. M. Barendregt[9], D. Pajkrt[9], M. van der Kuip[9], A. M. van Furth[9], Evelien Sprenkeler[25], Judith Zandstra[25], G. van Mierlo[25], J. Geissler[25], Dace Zavadska [32,33], Anda Balode[32,33], Arta Bārzdiņa[32,33], Dārta Deksne[32,33], Dace Gardovska[32,33], Dagne Grāvele[33], Ilze Grope[32,33], Anija Meiere[32,33], Ieva Nokalna[32,33], Jana Pavāre[32,33], Zanda Pučuka[32,33], Katrīna Selecka[32,33], Aleksandra Rudzāte[32,33], Dace Svile[33], Urzula Nora Urbāne[32,33], Nina A. Schweintzger [36], Werner Zenz[36], Benno Kohlmaier[36], Manfred G. Sagmeister[36], Daniela S. Kohlfürst[36], Christoph Zurl[36], Alexander Binder[36], Susanne Hösele[36], Manuel Leitner[36], Lena Pölz[36], Glorija Rajic[36], Sebastian Bauchinger[36], Hinrich Baumgart[69], Martin Benesch[70], Astrid Ceolotto[36], Ernst Eber[71], Siegfried Gallistl[36], Gunther Gores[72], Harald Haidl[35], Almuthe Hauer[36], Christa Hude[36], Markus Keldorfer[72], Larissa Krenn[73], Heidemarie Pilch[72], Andreas Pfleger[71], Klaus Pfurtscheller[73], Gudrun Nordberg[72], Tobias Niedrist[74], Siegfried Rödl[73], Andrea Skrabl-Baumgartner[36], Matthias Sperl[75], Laura Stampfer[72], Volker Strenger[70], Holger Till[69], Andreas Trobisch[72], Sabine Löffler[72], Ulrich von Both[10,11], Laura Kolberg [10], Manuela Zwerenz[10], Judith Buschbeck[10], Christoph Bidlingmaier[76], Vera Binder[77], Katharina Danhauser[78], Nikolaus Haas[79], Matthias Griese[80], Tobias Feuchtinger[77], Julia Keil[81], Matthias Kappler[80], Eberhard Lurz[82], Georg Muench[83], Karl Reiter[81], Carola Schoen[81], Ioanna Georgaki[23], Maria Tsolia[23], Irini Eleftheriou[23], Maria Tambouratzi[23], Antonis Marmarinos[23], Marietta Xagorari[23], Kelly Syggelou[23], Philipp K. A. Agyeman [7], Luregn J. Schlapbach[34,35], Christoph Aebi[7,84], Verena Wyss[7,84], Mariama Usman[7,84], Eric Giannoni[85,86], Martin Stocker[87], Klara M. Posfay-Barbe[88], Ulrich Heininger[89], Sara Bernhard-Stirnemann[90], Anita Niederer-Loher[91], Christian Kahlert[91], Giancarlo Natalucci[92], Christa Relly[93], Thomas Riedel[94], Christoph Berger[93], Mojca Kolnik[24], Marko Pokorn[24,30,31], Katarina Vincek[24], Tina Plankar Srovin[24], Natalija Bahovec[24], Petra Prunk[24], Veronika Osterman[24], Tanja Avramoska[24], Karen Brengel-Pesce[12,13], Marine Mommert-Tripon[12,13], François Mallet[12,13], Alexandre Pachot[12], Effua Usuf [40], Kalifa Bojang[40], Syed M. A. Zaman[40], Fatou Secka[40], Suzanne Anderson[40], Anna Roca[40], Isatou Sarr[40], Momodou Saidykhan[40], Saffiatou Darboe[40], Samba Ceesay[40], Umberto D'Alessandro[40], Ching-Fen Shen [37], Ching-Chuan Liu[37] & Shih-Min Wang[37]

[47]Department of Materials, Imperial College London, London, UK. [48]Imperial College Healthcare NHS Trust, London, UK. [49]Brighton and Sussex University Hospitals, Brighton, UK. [50]University of Liverpool, Liverpool, UK. [51]Alder Hey Children's Hospital, Liverpool, UK. [52]Department of Pathology, University of Otago, Christchurch, New Zealand. [53]Department of Paediatrics, University of Oxford, Oxford, UK. [54]Population Health Sciences Institute, Newcastle University, Newcastle upon Tyne, UK. [55]Newcastle University Business School, Centre for Knowledge, Innovation, Technology and Enterprise (KITE), Newcastle upon Tyne, UK. [56]Great North Children's Hospital Research Unit, Newcastle upon Tyne Hospitals NHS Foundation Trust, Newcastle upon Tyne, UK. [57]Great North Children's Hospital, Paediatric Intensive Care Unit, Newcastle upon Tyne Hospitals NHS Foundation Trust, Newcastle upon Tyne, UK. [58]Great North Children's Hospital, Paediatric Oncology, Newcastle upon Tyne Hospitals NHS Foundation Trust, Newcastle upon Tyne, UK. [59]Northumbria University, Newcastle upon Tyne, UK. [60]Unidade de Xenética, Departamento de Anatomía Patolóxica e Ciencias Forenses, Instituto de Ciencias Forenses, Facultade de Medicina, Universidade de Santiago de Compostela, Compostela, Spain. [61]GenPop Research Group, Instituto de Investigaciones Sanitarias (IDIS), Hospital Clínico Universitario de Santiago, Galicia, Spain. [62]Department of immunology, Erasmus MC-Sophia Children's Hospital, Rotterdam, the Netherlands. [63]Radboud University Medical Center, Nijmegen, the Netherlands. [64]Canisius Wilhelmina Hospital, Nijmegen, the Netherlands. [65]St. Antonius Hospital, Nieuwegein, the Netherlands. [66]Catharina Hospital, Eindhoven, the Netherlands. [67]ETZ Elisabeth, Tilburg, the Netherlands. [68]Franciscus Gasthuis, Rotterdam, the Netherlands. [69]Department of Paediatric and Adolescence Surgery, Medical University Graz, Graz, Austria. [70]Department of Pediatric

Hematooncoloy, Medical University of Graz, Graz, Austria. [71]Department of Pediatric Pulmonology, Medical University of Graz, Graz, Austria. [72]University Clinic of Paediatrics and Adolescent Medicine Graz, Medical University Graz, Graz, Austria. [73]Paediatric Intensive Care Unit, Medical University of Graz, Graz, Austria. [74]Clinical Institute of Medical and Chemical Laboratory Diagnostics, Medical University Graz, Graz, Austria. [75]Department of Pediatric Orthopedics, Medical University Graz, Graz, Austria. [76]Division of General Paediatrics, Hauner Children's Hospital, University Hospital, Ludwig Maximilians University (LMU), Munich, Germany. [77]Division of Paediatric Haematology & Oncology, Hauner Children's Hospital, University Hospital, Ludwig Maximilians University (LMU), Munich, Germany. [78]Division of Paediatric Rheumatology, Hauner Children's Hospital, University Hospital, Ludwig Maximilians University (LMU), Munich, Germany. [79]Department Pediatric Cardiology and Pediatric Intensive Care, University Hospital, LMU, Munich, Germany. [80]Division of Paediatric Pulmonology, Hauner Children's Hospital, University Hospital, Ludwig Maximilians University (LMU), Munich, Germany. [81]Paediatric Intensive Care Unit, Hauner Children's Hospital, University Hospital, Ludwig Maximilians University (LMU), Munich, Germany. [82]Division of Paediatric Gastroenterology, Hauner Children's Hospital, University Hospital, Ludwig Maximilians University (LMU), Munich, Germany. [83]Neonatal Intensive Care Unit, Hauner Children's Hospital, University Hospital, Ludwig Maximilians University (LMU), Munich, Germany. [84]Department of Pediatrics, lnselspital, Bern University Hospital, University of Bern, Bern, Switzerland. [85]Clinic of Neonatology, Department Mother-Woman-Child, Lausanne University Hospital and University of Lausanne, Lausanne, Switzerland. [86]Infectious Diseases Service, Department of Medicine, Lausanne University Hospital and University of Lausanne, Lausanne, Switzerland. [87]Department of Pediatrics, Children's Hospital Lucerne, Lucerne, Switzerland. [88]Pediatric Infectious Diseases Unit, Children's Hospital of Geneva, University Hospitals of Geneva, Geneva, Switzerland. [89]Infectious Diseases and Vaccinology, University of Basel Children's Hospital, Basel, Switzerland. [90]Children's Hospital Aarau, Aarau, Switzerland. [91]Division of Infectious Diseases and Hospital Epidemiology, Children's Hospital of Eastern Switzerland St. Gallen, St. Gallen, Switzerland. [92]Department of Neonatology, University Hospital Zurich, Zurich, Switzerland. [93]Division of Infectious Diseases and Hospital Epidemiology, and Children's Research Center, University Children's Hospital Zurich, Zurich, Switzerland. [94]Children's Hospital Chur, Chur, Switzerland.

