## [Transparent Peer Review file · Nature Communications]

Predicting trajectories of illness using RNA velocity of whole blood

Corresponding Author: Professor Aubrey Cunnington

Version 0:

Reviewer comments:

Reviewer #1

(Remarks to the Author)

In this manuscript Dunican and colleagues adapted RNA velocity, an analytical tool commonly used in single cell RNAseq that includes unspliced and spliced mRNA of each cell, to predict future clinical status of an individual patient by leveraging their whole blood transcriptome analysis. They evaluated this approach, that they named VeloCD, to predict future transcriptomic and clinical status in three different settings: 1) human challenge studies with influenza A and SARS-CoV-2; 2) a cohort of HIV-TB coinfecting individuals ; and 3) in a cohort of febrile children.

This is an interesting study that provides an alternative methodological approach to use RNA velocity in a more clinically relevant context. There are important aspects that deserve clarification and/or additional information.

Specific Comments

1. CHIMS studies, did the authors observe any changes in gene expression either unspliced, spliced or both before the PCR became positive?
2. It is not clear how IFI44L and LY6E were selected to show the dynamic changes of transcriptomes. Are they the top 2 genes for both cohorts based on maSigPro analysis? It is a little surprising that there are no obvious differences in terms of expression at each time point for spliced and un-spliced transcripts of these two genes when comparing Fig 1 and Fig 2.
3. Did the VeloCD predict clinical phenotypes in the CHIM studies?
4. For prediction of TB-IRIS, only the sensitivity was described with different threshold, it would be ideal to also add specificity (false positive) as well for each test. The authors may consider using only randomly selected half of no-TB-IRIS as input to VeloCD and add the other half as the test group.
5. In the PERFORM study, most of mild and severe cases can be classified correctly using VeloCD. However, it is not clear if RNA velocity prediction model performs better than traditional prediction model based on gene expression values. In this context, it would be ideal if the authors could compare VeloCD with more established methods using traditional gene expression analyses.

(Remarks on code availability)

It is adequate and very well organized.

Reviewer #2

(Remarks to the Author)

Thank you for the opportunity to review this interesting study. The authors present the development and testing of a new method of predicting future clinical states based on RNA measurements taken at an earlier time point. The new methodology, VeloCD, is based on measuring RNA velocity, which in turn relies on comparing spliced and unspliced transcripts at a single time point in order to predict the spliced transcriptome at a later time point. The authors demonstrate the prediction of future RNA and clinical state in a small cohort of healthy volunteers inoculated with influenza or SARS-CoV-2 virus. They then test the method with single-timepoint RNA data from a medium-sized cohort of patients with HIV-TB coinfection, and a larger cohort of pediatric patients with infectious conditions.

Overall the new method extends prior work on RNA velocity done in single-cell models, and explores the utility in a clinical application. It benefits from the use of 3 distinct datasets of varying size and detail. The method described could potentially be a bridge towards clinical applications of this methodology, however some hurdles remain and there are findings that will have to be confirmed in larger, prospectively collected datasets.

Primary questions:

1. It appears that the RNA velocity concept is established for single-cell RNA Seq but its validity in bulk RNA sequencing experiments such as the ones used here is less certain. The reader would benefit from some additional exposition on this point. How does the method accommodate for the fact that multiple different cell types will be captured in the RNA data? Was the sequencing adequate to measure the necessary unspliced RNA? Was the RNA extracted from the clinical samples of adequate quality? What did the authors do to account for batch effects (PMID 39696422)? Are there any technical manuscripts or white papers on measuring RNA velocity in these types of whole blood samples?
2. An important consideration seems to be that the future clinical state of interest must be anticipated. In real-world deployment this is unknown at the time of clinical decision-making, and in fact sometimes even the baseline state is unknown. The drop-one-in method seems to require the user to select neighbours with a certain outcome (eg. influenza PCR positive), but if all you know is that a patient has fever, how do you know which of these future states to include in the model? In general, it seems like the application here is to answer a question like "Patient A was exposed to influenza virus – how likely are they to get infected?", which has less clinical utility overall.
3. There may be an overfitting issue, as evidenced by the drop in performance when predictor transcripts for flu were used for COVID prediction, and vice versa. There is a decrement in performance (understandably) but this "mismatch" performance might be a more accurate approximation of eventual clinical deployment.
4. While this may represent a technical advance in modeling, the higher-impact use case is in the potential applications. If the aim is to predict future clinical states, it would be helpful to see how this approach compares to other (simpler) prediction models. Could a clinical prediction model (derived from baseline clinical parameters) perform as well? Additionally, the authors note "In many cases, small sets of genes ("signatures") have been identified to discriminate between different current disease states." Can these signatures discriminate between different outcome states as well? Such models would presumably be simpler, and might help answer the question of whether it is truly the RNA velocity that is predictive, or whether it is simply the high-dimensional transcriptomic data that is contributing the majority of the discrimination. In general, it would help the reader to understand if there is added value to this approach over simpler methods. One benefit is potential biological insights, however these might vary from one use case to another.

Minor issues/areas for improvement:

5. In general, the Figures are helpful but some improvements could be made, mostly because the individual panels are often quite small and difficult to view. Could these be broken up into more individual figures?
6. Fig 1f – missing symbols in the legend for PCR status
7. Fig 1e and 1f are quite small and there's a lot of overlap. It's also noteworthy that the first two PCs don't explain all that much variation in the data. Perhaps the final PC plots ("i" and "j") are sufficient?
8. Fig 1g and 1h – what do the various colours in the volcano plots represent? I presume that red dots indicate transcripts that are above a certain fold-change threshold and below a corrected p-value threshold. Not clear what the utility of the other colours is.
9. Fig 1k and 1l don't have legends for the colour and shape of symbols (though it's fairly clear from the context)
10. Fig 2h symbol is missing in the legend for PCR status
11. How are Fig 2g and 2h to be interpreted? How do we know the trajectories indicated reflect the actual RNA dynamics? Perhaps a correlation plot (or series of correlation plots) like Fig 2e/f for more of the downstream time points would be helpful.
12. How did you choose IFI44L and LY6E from among all the transcripts? Is this a manual selection process or something that is automated/algorithmic within the VeloCD method?
13. It appears the "future" state of spliced RNA is only 1 day ahead, which depending on time of day of the sampling might be a relatively short time (in terms of illness progression and clinical utility).
14. Fig 3 – the ROC curves likely reflect overfitting based on the nature of the data and the derivation/validation used.
15. "Interestingly, applying this signature to the day 0 (post-inoculation) samples, yielded a predictive performance similar to when day 1 samples were used as the test set (optimal AUROC, 0.73, 95% CI: 0.51-0.95, TPNN: 10, full range of AUROCs: 0.52-0.73), indicating potential to predict up to 3 days into the future." Did the authors look at using Day 0 pre-inoculation samples? If these can generate accurate predictions there may be some very interesting implications.

(Remarks on code availability)

Reviewer #3

(Remarks to the Author)

This manuscript presents a novel application of RNA velocity, a technique typically used in single-cell analysis, to predict the future clinical trajectory of acute illness using whole-blood transcriptomics. The authors developed a new tool, VeloCD, for this purpose and applied it to three diverse datasets: controlled human infection models (CHIMs) for influenza and SARS-CoV-2, a clinical trial for TB-associated immune reconstitution inflammatory syndrome (TB-IRIS), and a large observational study of febrile children (PERFORM). Compact gene signatures (as small as three transcripts) achieve AUROC 0.72–0.89 for near term infection status and identify high risk children who later require PICU. The core strength is the innovative adaptation of RNA velocity from single-cell analysis to bulk whole-blood transcriptomics for predicting clinical trajectories. This opens a new avenue for utilizing transcriptomic data in prognosis.

Major concerns

1. Internal reuse of data for discovery and testing likely inflates AUROC and obscures generalizability. The authors should hold out one full cohort (e.g., influenza) for blind evaluation or deploy the three gene influenza panel on an external ARI dataset such as GSE73072 without re training.
2. Velocity fields are built from very small or imbalanced reference groups, raising stability concerns. The authors should perform a learning curve analysis that subsamples reference sets to quantify the minimum sample size needed for robust performance and report that threshold in the Methods.
3. PCA plots suggest batch effects and heterogeneous library chemistries that may bias velocity estimates. The authors should apply a bulk batch correction method before velocity modelling, document the procedure, and display corrected PCA/UMAP plots to demonstrate removal of technical variance.
4. Selection of the Transition Probability Neighbor Number (TPNN) appears ad hoc and hampers reproducibility. The authors should implement an automated grid search or cross validation routine within VeloCD that chooses TPNN objectively and describe the default optimization rule in the user guide.
5. Clinical interventions in the PERFORM cohort confound the link between predicted and true trajectories. The authors should restrict evaluation to outcomes measured before major treatments or adjust for treatment probability using causal inference techniques such as inverse probability weighting.
6. The manuscript offers little mechanistic explanation for why the selected genes predict deterioration. The authors should run GOBP/KEGG enrichment for each final signature and relate velocity directions to known immune pathways, discussing how genes like BAK1 and APOL3 fit those pathways.
7. Predictive value is not benchmarked against standard clinical scores or existing expression panels. The authors should compare VeloCD probabilities with NEWS2/qSOFA (adults), PEWS (children), CRP levels, and the 2 gene bacterial/viral score, presenting ROC or decision curve analyses and net reclassification statistics.
8. Generalizability beyond infectious or immune mediated conditions remains untested. The authors should evaluate VeloCD on at least one non infectious dataset or explicitly discuss current limitations and a roadmap for extending the approach to chronic and non infectious diseases.

Minor concerns

1. Figure resolution and legends not publication ready. The authors should increase DPI for all figures, adopt color blind safe palettes, and restore the missing PCR status symbol in Fig 1f.
2. Text discrepancies create confusion. The authors should change “five genes” to “three genes” (line 243), replace “focussed” with “focused” (line 111), and run a full spell check.
3. Separate spliced and unspliced plots obscure kinetic parallels. The authors should overlay the traces (e.g., combine Fig 1k and Fig 2a) to better illustrate matching expression dynamics.
4. Velocity arrows in fate maps are qualitative and lack quantitative backing. The authors should provide numeric summaries and consider adding a “group to group velocity” statistic to VeloCD.

(Remarks on code availability)

Version 1:

Reviewer comments:

Reviewer #2

(Remarks to the Author)

Thank you for the careful responses and additional work presented. No further comments.

(Remarks on code availability)

Reviewer #3

(Remarks to the Author)

The authors have made substantial efforts to address all my concerns if possible, particularly regarding generalizability, batch effects, and mechanistic explanations. I have no more comments.

(Remarks on code availability)

Predicting trajectories of illness using RNA velocity of whole blood

Reviewer	Reviewer Comment	Response	Changes Made
#1	1. CHIMS studies, did the authors observe any changes in gene expression either unspliced, spliced or both before the PCR became positive?	Yes, we do observe changes in gene expression before PCR becomes positive, as shown for specific genes in Figure 1 k and l, Figure 2 a-d and Supplementary Figures 1 and 5.	We have added a sentence to the main text to highlight this. Line 245 "These figures also demonstrate that the changes in spliced and unspliced transcript expression are apparent before individuals test positive by PCR, which is also evident in the corresponding phase portraits (Supplementary Fig. 1)."
	2. It is not clear how IFI44L and LY6E were selected to show the dynamic changes of transcriptomes. Are they the top 2 genes for both cohorts based on maSigPro analysis? It is a little surprising that there are no obvious differences in terms of expression at each time point for spliced and un-spliced transcripts of these two genes when comparing Fig 1 and Fig 2.	IFI44L and Ly6E were selected for illustration because they were in common between the maSigPro analyses of the two CHIM datasets. Figure 1k,l and Figure 2a-d are not intended to illustrate the changes in spliced and unspliced transcript expression at the level of individual subjects, they illustrate the parallel trajectories over the whole set of subjects. To appreciate the changes on an individual basis, which underlie the calculations of RNA velocity, one needs to look at the phase portraits (spliced vs unspliced expression) for individual subjects.	We have amended the main text to explain more clearly how IFI44L and LY6E were selected. Line 229: "Two of the top five genes (ordered by multiple testing corrected p-value) were shared across these analyses..." We have added a new Supplementary Figure 1 illustrating phase portraits for IFI44L and Ly6E.
	3. Did the VeloCD predict clinical phenotypes in the CHIM studies?	Thank you for this interesting question. The available indicator of clinical phenotypes in the CHIM studies is symptom score, which we have available for the influenza CHIM. The symptom scores do indeed correlate with the predictions of becoming PCR positive using VeloCD. We have added this to the text.	We have added this to the main text along with a new supplementary figure, Line 344: "We investigated if these probabilities correlated with future symptom scores for the study participants. Using day 1 samples, the VeloCD-predicted probabilities of becoming PCR positive (by the end of the study), were moderately correlated with symptom score at day 2 (p-value = 0.023, Pearson correlation coefficient: 0.47, n=23 – of which 14 had non-zero symptom scores, Supplementary Fig. 7)."
	4. For prediction of TB-IRIS, only the sensitivity was described with different threshold, it would be ideal to also add specificity (false positive) as well for each test. The authors may consider using only randomly selected half of no-TB-IRIS as input to VeloCD and add the other half as the test group.	We were reluctant to perform this analysis in our original manuscript because of the small number of TB-IRIS events and the imbalanced number of subjects with TB-IRIS vs no TB-IRIS. However, we have now added this as suggested by the reviewer. The results indicate a modest performance for prediction of TB-IRIS.	We have added this new analysis to the main text along with a new supplementary figure, Line 468: "To formally assess how well RNA velocity predicts the onset TB-IRIS in this dataset, we created a test set from 25 randomly selected subjects who did not develop TB-IRIS and

			the eight subjects who developed TB-IRIS after day 14, and re-ran analysis, selecting a new 56-gene signature (Supplementary Figure 14..."
	5. In the PERFORM study, most of mild and severe cases can be classified correctly using VeloCD. However, it is not clear if RNA velocity prediction model performs better than traditional prediction model based on gene expression values. In this context, it would be ideal if the authors could compare VeloCD with more established methods using traditional gene expression analyses.	Thank you for this suggestion. We have added a comparison with a more traditional prediction model based on "static" gene expression. The analysis is under-powered to formally demonstrate superiority of VeloCD vs the static gene expression model for prediction of outcome, but there is suggestive evidence that VeloCD augments prediction beyond that of the static gene expression model.	We have added new analysis to the main text along with new supplementary figures 18e and 19. Line 655: "We also used the PERFORM dataset to compare RNA velocity-based predictions with predictions from a "static" gene expression signature ..."
#2	1. It appears that the RNA velocity concept is established for single-cell RNA Seq but its validity in bulk RNA sequencing experiments such as the ones used here is less certain. The reader would benefit from some additional exposition on this point. How does the method accommodate for the fact that multiple different cell types will be captured in the RNA data? Was the sequencing adequate to measure the necessary unspliced RNA? Was the RNA extracted from the clinical samples of adequate quality?	The first paper describing the concept of RNA velocity analysis (La Manno et al., Nature 2018) did focus primarily on single-cell RNA seq data but also provided evidence for application to bulk RNA sequence data, predicting circadian gene expression in liver tissue. The application of RNA velocity to bulk whole blood RNA-seq data is one of the novelties of our approach. The method does not attempt to account for multiple cell types contributing to the bulk RNA data. This is because many publications have shown that robust gene expression signatures, derived from whole blood bulk RNA-seq data and intended for diagnostic or prognostic use, can be discovered without the need to account for cell mixture. Therefore, we hypothesized that it may be possible to identify sets of genes suitable for RNA velocity analysis, for which this would also be the case. This would greatly simplify clinical translation, because there would be no need to have additional quantification of cell mixtures. We are not aware of any formal criteria to assess adequacy of sequencing depth for quantification of unspliced RNA, but intronic reads constituted around half of the uniquely mapped reads in the majority of datasets (Supplementary Table 1) We do not have data on RNA integrity for all studies, although they reported that it was assessed and we assume only adequate quality samples were taken forward for sequencing. In the PERFORM dataset, due to the large number of subjects, every 4th sample was assessed, and in every case the RIN was at least 7.	We have incorporated this into the introduction, Line 130, "It was also demonstrated that RNA velocity analysis could be applied to bulk RNA-seq of mouse liver tissue to model circadian changes in gene expression." We have also re-emphasized the fact that we are using bulk RNA-seq data throughout the manuscript to avoid any confusion. We have incorporated this point into the introduction, Line 138: "Despite blood having a complex cellular composition, which can change during illness, we and others have demonstrated that it is possible to find transcriptomic signatures in whole blood bulk RNA, which robustly identify different causes and states of acute illness without needing to adjust for cell mixture..." Supplementary Table 1 has been updated to show intronic and exonic read depths.

	What did the authors do to account for batch effects (PMID 39696422)? Are there any technical manuscripts or white papers on measuring RNA velocity in these types of whole blood samples?	Potential batch effects were explored in all datasets, by examining known potential sources of technical variation on principal component plots. We did not identify any evidence for technical batch effects. We note that PMID39696422 refers to a single cell RNA-seq analysis – this method is more prone to batch effects. We used bulk RNA-seq as the basis for our analyses. Please also see response to Reviewer 3 comment 3, and figure for review only below. We are not aware of any technical manuscripts or white papers on measuring RNA velocity on whole blood bulk RNA-seq samples. We believe that our study provides the first proof of concept of this approach.	We have added explanation of how we assessed batch effects to the methods section, Line 962: “We used PCA plots to exclude potential batch effects associated with known technical factors in each dataset.”
	2. An important consideration seems to be that the future clinical state of interest must be anticipated. In real-world deployment this is unknown at the time of clinical decision-making, and in fact sometimes even the baseline state is unknown. The drop-one-in method seems to require the user to select neighbours with a certain outcome (eg. influenza PCR positive), but if all you know is that a patient has fever, how do you know which of these future states to include in the model? In general, it seems like the application here is to answer a question like “Patient A was exposed to influenza virus – how likely are they to get infected?”, which has less clinical utility overall.	This is an important point and is the reason we have included the PERFORM dataset, which allows use to explore a realistic clinical application. The reviewer is absolutely correct that predictions using RNA velocity must be made with reference to gene expression in known outcome groups, but these reference groups can be constructed so as to be representative of the population in which the test would be performed. In the PERFORM dataset, we aimed to predict trajectory of illness towards either end of spectrum of severity – one reference group being critical illness and the other mild illness. Based on the clinical experience of many of the authors, a test which could predict trajectory of illness in patients who appear moderately ill at the time of assessment could be incredibly useful to assist with decisions about which patients should be admitted to hospital and which can be managed safely at home. The CHIM studies are presented because the controlled conditions and serial samples allow us to demonstrate that application of RNA velocity to whole blood meets all of the requirements for the method to be valid. We don’t consider that predicting who will become infected with influenza or SARS-CoV-2 will be game changing, although we suggest that this could be explored as an important application for high consequence infectious diseases (eg viral hemorrhagic fevers).	We have added further explanation about the relevance of the analysis of the PERFORM dataset, Line 532: “This dataset therefore captures the real-world clinical scenario of patients presenting to emergency departments and clinical decisions about hospital admission and treatment needing to be made. “ We have also added further explanation about the potential application of the method to predicting infection following exposure, Line 381: “A potential application could be early identification and isolation of individuals who have been infected following exposure to high consequence pathogens, before the pathogen itself becomes detectable or transmissible to others.”

3. There may be an overfitting issue, as evidenced by the drop in performance when predictor transcripts for flu were used for COVID prediction, and vice versa. There is a decrement in performance (understandably) but this “mismatch” performance might be a more accurate approximation of eventual clinical deployment.	We agree with the reviewer and have already alluded to this in the discussion, Line 733: “...our prediction of future disease states was made using selected subsets of genes and hyperparameters, which risks over-optimistic performance”. However, there are also other factors which might impact on generalizability, like reference set size and composition (please also see response to Reviewer 3). We consider that the “signatures” we have derived in this study provide proof of principle that RNA velocity can be used to make predictions of clinical states, but these are not intended to be optimal, generalizable signatures, which would need to be discovered and validated with much larger datasets.	We have added to the discussion: Line 729: “...we found evidence that the size and composition of the reference data set can impact predictions.” Line, 744: “The gene signatures selected in this study provide examples of potential utility but are not intended as definitive signatures for potential clinical use-cases.”
4. While this may represent a technical advance in modeling, the higher-impact use case is in the potential applications. If the aim is to predict future clinical states, it would be helpful to see how this approach compares to other (simpler) prediction models. Could a clinical prediction model (derived from baseline clinical parameters) perform as well? Additionally, the authors note ““In many cases, small sets of genes (“signatures”) have been identified to discriminate between different current disease states.” Can these signatures discriminate between different outcome states as well? Such models would presumably be simpler, and might help answer the question of whether it is truly the RNA velocity that is predictive, or whether it is simply the high-dimensional transcriptomic data that is contributing the majority of the discrimination. In general, it would help the reader to understand if there is added value to this approach over simpler methods. One benefit is potential biological insights, however these might vary from one use case to another.	Thank you for this suggestion. We have assessed this using the PERFORM dataset. We considered a variety of currently available clinical risk stratification tools, but the available clinical data was only sufficiently complete to enable us to use the National Institute of Health and Care Excellence (NICE) criteria for stratification of risk of severe illness or death from sepsis (which is a widely used tool in the UK). We compared this with predictions made using VeloCD. This analysis shows that RNA velocity adds more useful information than the NICE risk stratification tool. Thank you for this question, which is similar to Reviewer 1 Comment 5. We have addressed this in response to Reviewer 1. There does indeed seem to be benefit of including RNA-velocity beyond that of just using high dimensional transcriptomic data.	We have added to the results, Line 640: “We were interested to compare the predictions of future severity using VeloCD with risk stratification based on clinical data...” and Supplementary Figure 18d. We have added further details to the Methods, Line 869: “The NICE criteria for stratification of risk of severe illness or death from sepsis were used to categorize risk of serious illness according to clinical features...” Please see response to Reviewer 1 comment 5. We included references to illustrate that disease outcomes can be predicted by gene signatures (Line 119)
5. In general, the Figures are helpful but some improvements could be made, mostly because the individual panels are often quite small and difficult to view. Could these be broken up into more individual figures?	We have received re-formatting instructions from the editor and hope that you will find the new figures clearer. We have tried to retain related panels within the same figure because we believe that ultimately, in a final formatted publication, this will be easier to read.	All figures have been revised according to journal formatting instructions.
6. Fig 1f – missing symbols in the legend for PCR status	Thank you.	Corrected
7. Fig 1e and 1f are quite small and there’s a lot of overlap. It’s also noteworthy that the	Figures 1e and 1f are required to show poorer separation before DEA (as a before	Plots have been retained in final figures.

	first two PCs don't explain all that much variation in the data. Perhaps the final PC plots ("i" and "j") are sufficient?	and after contrast) and justify the choice of the days for DEA. The modest variance explained by PC1 and PC2 in these plots is a result of all expressed genes being used to construct the PCs, whereas 1i and 1j show PCA plots based only on differentially expressed genes (inevitably PC1 and PC2 now account for more of the variation)	
	8. Fig 1g and 1h – what do the various colours in the volcano plots represent? I presume that red dots indicate transcripts that are above a certain fold-change threshold and below a corrected p-value threshold. Not clear what the utility of the other colours is.	We apologise that this was not clear. The colours represent significance and log-fold change.	Figure legend has been amended to explain the meaning of the colours.
	9. Fig 1k and 1l don't have legends for the colour and shape of symbols (though it's fairly clear from the context)	Thank you for highlighting this.	Figure has been updated with legend
	10. Fig 2h symbol is missing in the legend for PCR status	Thank you for highlighting this.	Figure legend has been corrected
	11. How are Fig 2g and 2h to be interpreted? How do we know the trajectories indicated reflect the actual RNA dynamics? Perhaps a correlation plot (or series of correlation plots) like Fig 2e/f for more of the downstream time points would be helpful.	The trajectories indicated by the velocity arrows do indeed appear to explain RNA dynamics because the arrows are concordant with temporal progression of samples in transcriptomic space. This would be a great way to illustrate the point if we did not have gaps in sampling (influenza CHIM samples were only collected on days 0,1,2,3,7,10). However, we have added an additional set of correlation plots to address your later comment (13) which provides further supporting evidence.	We have added additional explanation to the text, Line 277: "Arrows on these fate maps represent the magnitude and direction of RNA velocity for each sample across all relevant genes in the depicted transcriptomic space. The RNA velocities are highly concordant with the observed temporal progression of the transcriptome along diverging trajectories for infected and uninfected subjects." Please see response to your comment 13.
	12. How did you choose IFI44L and LY6E from among all the transcripts? Is this a manual selection process or something that is automated/algorithmic within the VeloCD method?	Please see response to Reviewer 1 comment 2. This is a manual step, performed to illustrate that RNA dynamics can be measured in the CHIM datasets, establishing suitability for application of RNA velocity analysis to samples from individuals rather than single cells.	Please see response to Reviewer 1 comment 2.
	13. It appears the "future" state of spliced RNA is only 1 day ahead, which depending on time of day of the sampling might be a relatively short time (in terms of illness progression and clinical utility).	Thank you for raising this point. We can only investigate this in the serial samples from the CHIM datasets, where we find that strong correlations between predicted and measured spliced transcript expression can be observed over longer time periods: we used day three samples to predict expression at day seven. This is also consistent with the VeloCD predictions in the TB-IRIS and newly added IBD data sets, which predict days or weeks into the future.	We have added a new Supplementary Figure 2 to demonstrate this, and have added explanatory text, Line 269. "In the influenza dataset, future spliced transcript expression predicted at day 3 was also highly correlated with the actual expression at day 7 (Pearson correlation coefficients, 0.86-0.99; mean, 0.94 between

			subject samples; Supplementary Fig. 2a)..."
	14. Fig 3 – the ROC curves likely reflect overfitting based on the nature of the data and the derivation/validation used.	Please see response to your comment 3	Please see response to your comment 3
	15. "Interestingly, applying this signature to the day 0 (post-inoculation) samples, yielded a predictive performance similar to when day 1 samples were used as the test set (optimal AUROC, 0.73, 95% CI: 0.51-0.95, TPNN: 10, full range of AUROCs: 0.52-0.73), indicating potential to predict up to 3 days into the future. " Did the authors look at using Day 0 pre-inoculation samples? If these can generate accurate predictions there may be some very interesting implications.	We have looked at whether there was any evidence of difference in gene expression at Day 0, pre-inoculation, between those who became infected and those who did not become infected – there were no differentially expressed genes. Since subjects were not exposed to the pathogen before inoculation, we have not looked at RNA velocity predictions using the pre-inoculation samples, but draw the reviewers attention to Fig 2h where the RNA velocities of these samples are seen to be negligible, consistent with their healthy unexposed state.	Added to the results section, Line 211: "... (there were no differentially expressed genes prior to virus inoculation between subjects who became infected or remained uninfected)."
#3	1. Internal reuse of data for discovery and testing likely inflates AUROC and obscures generalizability. The authors should hold out one full cohort (e.g., influenza) for blind evaluation or deploy the three gene influenza panel on an external ARI dataset such as GSE73072 without re training.	We agree with the reviewer that it would be ideal to validate all of the predictive signatures in independent datasets. We would like to clarify that the influenza and SARS-CoV-2 CHIM studies were conducted independently, and so testing the predictive RNA velocity signature from the influenza dataset on the SARS-CoV-2 dataset does constitute an independent validation. We also note that we tested this signature directly on this dataset, without any reweighting, to provide a realistic assessment of its performance. We do agree that it would be ideal if we could test these signatures for prediction of infection state in other ARI datasets, but we conducted an exhaustive search for other suitable datasets and could not identify any. The suggested GSE73072 is unfortunately a microarray dataset, which is not suitable for RNA velocity analysis. To provide further evidence of generalizability of trajectories indicated by RNA velocity from CHIM to naturally acquired infection, we have added an analysis data from the Integrated Network for Surveillance, Trials and Investigations into COVID-19 Transmission (INSTINCT) study. This dataset did not include pre-detection samples from individuals who later became PCR positive, so it was not possible to assess the performance of the predictive signatures. Similar to our response to Reviewer 2 Comment 3, we would like to clarify that the the gene signatures selected in this study provide proof-of-concept and examples of potential utility but are not intended as definitive signatures for potential clinical use-cases. We believe that our existing discussion already adequately addresses the need for validation in independent datasets.	We have clarified in the introduction that RNA velocity uses RNA-seq data, Line 127-132. We have added a new analysis of the INSTINCT cohort, with new Supplementary Figure 11 and described from Line 385: "Generalizability of RNA velocity to natural SARS-CoV-2 infection..." and methods, Line 792 onwards. We have added to the Discussion, Line, 744. "The gene signatures selected in this study provide examples of potential utility but are not intended as definitive signatures for potential clinical use-cases." We have also amended the abstract to clarify that we consider our findings to be "proof-of-concept".

	2. Velocity fields are built from very small or imbalanced reference groups, raising stability concerns. The authors should perform a learning curve analysis that subsamples reference sets to quantify the minimum sample size needed for robust performance and report that threshold in the Methods	Thank you for this suggestion. Using the largest dataset (PERFORM) we have assessed the impact of varying reference group size, reference group composition, and the number of neighbours (NN) parameter (which determines the spatial organization of the reference groups) to provide insights into the effect of each parameter on predictions made by VeloCD. We have not depicted this through classical learning curves, because we need to illustrate the effect of multiple parameters which influence transition probabilities (reference size, reference composition, NN, and TPNN values) but we believe the findings are informative because they illustrate the importance of reference size and composition. These findings will inform future refinements to the methodology.	We have added Supplementary Figure 20 and a description of this analysis at Line 673: "Finally, we took advantage of this relatively large data set to examine the effect of varying reference set size on stability of VeloCD TPs..."
	3. PCA plots suggest batch effects and heterogeneous library chemistries that may bias velocity estimates. The authors should apply a bulk batch correction method before velocity modelling, document the procedure, and display corrected PCA/UMAP plots to demonstrate removal of technical variance.	We found this comment a little confusing because we did not display any PCA plots illustrating technical factors which may cause bias, and we are unsure whether this comment refers to one or more of the datasets presented in the manuscript. As part of our normal quality control and exploratory analysis procedures we examine the effect of "batch" in low dimensional transcriptomic space (via PCA). Batch effects can occur as result of many technical factors such as plate, extraction, cleanup, and library preparation method. We examined these systematically using the information available to us relating to each of RNA-Seq datasets. We provide a figure at the end of this response document, "For Review Only", illustrating our analyses for batch effects. These show no evidence of batch effects which are likely to have significant impacts on our analyses. The plots shown were generated using the biplot method, with the shaded areas generated using the encircle method (for each "batch" group).	We have added explanation of how we assessed batch effects to the methods section, Line 962: "We used PCA plots to exclude potential batch effects associated with known technical factors in each dataset."
	4. Selection of the Transition Probability Neighbor Number (TPNN) appears ad hoc and hampers reproducibility. The authors should implement an automated grid search or cross validation routine within VeloCD that chooses TPNN objectively and describe the default optimization rule in the user guide.	We have updated the code on GitHub to automatically report the optimal TPNN for each analysis.	Code updated.
	5. Clinical interventions in the PERFORM cohort confound the link between predicted and true trajectories. The authors should restrict evaluation to outcomes measured before major treatments or adjust for treatment probability using causal inference techniques such as inverse probability weighting.	Unfortunately, we are not able to perform the suggested analysis because we do not have the necessary granularity of detail about timing, sequence, and effect of interventions in this dataset. In our experience, the majority of "major treatments" in the moderate illness group, such as administration of antibiotics, intravenous fluid boluses, and oxygen, are initiated in the Emergency Department within the first few hours of presentation. Despite not being able to undertake the suggested analysis, we would like to draw the reviewer's attention to the additional analysis we have undertaken using the	Please see response to Reviewer 2 comment 4.

		National Institute of Health and Care Excellence (NICE) criteria for stratification of risk of severe illness or death from sepsis (Reviewer 2 comment 4), which provides further context for assessment of trajectories. We note that this limitation was addressed in the discussion section of our original manuscript submission, now Line 736 onwards.	
	6. The manuscript offers little mechanistic explanation for why the selected genes predict deterioration. The authors should run GOBP/KEGG enrichment for each final signature and relate velocity directions to known immune pathways, discussing how genes like BAK1 and APOL3 fit those pathways.	Although it was not our intention to undertake any mechanistic exploration as part of this manuscript, we have performed GO analyses for the selected signatures. One important caveat is that we have selected genes based solely on their contributions to prediction of the outcomes of interest, without consideration of their biological context, and so they may provide an incomplete picture of mechanisms.	We have added GO analyses for gene signatures in Supplementary File 1, and have mentioned function of the "top" genes and GO terms for each signature in the relevant place in the Results section, e.g. Line 350: "The functions of these genes were assessed using gene ontology (GO) analysis, identifying 241 significant terms (after multiple testing correction) including "defense response to viruses" and "response to other organism "... We have added explanation of the GO analysis in the methods section, Line 1007
	7. Predictive value is not benchmarked against standard clinical scores or existing expression panels. The authors should compare VeloCD probabilities with NEWS2/qSOFA (adults), PEWS (children), CRP levels, and the 2 gene bacterial/viral score, presenting ROC or decision curve analyses and net reclassification statistics.	Thank you for this comment which is similar to Reviewer 2 Comment 4. The only existing clinical "score" which could be calculated for the majority of subjects in our dataset was the National Institute of Health and Care Excellence (NICE) criteria for stratification of risk of severe illness or death from sepsis. We have compared VeloCD predictions with this classification. We have undertaken additional analysis using CRP as a predictor of outcome, but there is an important caveat for this analysis, that CRP is used in the diagnostic classification algorithm and therefore is not independent of the selection of subjects for the VeloCD analyses of the PERFORM dataset. The 2-gene bacterial/viral score (which we assume refers to the Herberg et al. 2016 paper, DOI 10.1001/jama.2016.11236) is a diagnostic rather than prognostic signature. Rather than assessing the performance of this 2-gene signature for prediction, we derived a new static gene expression signature to predict outcome and assessed	Please see response to Reviewer 2 Comment 4. We have added analysis of CRP as a predictor of outcome, Line 632 "Next, we evaluated the performance of CRP (measured at the time of RNA sample collection) for distinguishing those who later transitioned to the PICU..." Please see response to Reviewer 1 comment 5.

		this in comparison to VeloCD (see Reviewer 1 comment 5) We have presented ROC curves for each of these analyses. It is important to note that the gene signatures selected in this study provide examples of potential utility but are not intended as definitive signatures for potential clinical use-cases.	We have noted this in the discussion, Line 744. “The gene signatures selected in this study provide examples of potential utility but are not intended as definitive signatures for potential clinical use-cases.”
	8. Generalizability beyond infectious or immune mediated conditions remains untested. The authors should evaluate VeloCD on at least one non-infectious dataset or explicitly discuss current limitations and a roadmap for extending the approach to chronic and non infectious diseases.	Thank you for this suggestion. It was challenging to identify a dataset from a non-infectious disease with RNA-seq data available for suitable reference outcome groups and a suitable test group with available RNA-seq data collected prior to known future outcomes. We identified one small study of subjects with inflammatory bowel disease with sampling during their first course of treatment with anti-TNF therapy and used this for VeloCD analysis to predict whether treatment resulted in remission. We have included this analysis as a new section in the manuscript.	New analysis added, Supplementary figure 15, and text starting line 482: “RNA velocity predicts the response after treatment for IBD...” New text added to methods, Line 812 onwards.
	Minor 1. Figure resolution and legends not publication ready. The authors should increase DPI for all figures, adopt color blind safe palettes, and restore the missing PCR status symbol in Fig 1f.	All figures have been updated according to Editorial guidelines, with Figure legends corrected where necessary.	All figures have been updated
	Minor 2. Text discrepancies create confusion. The authors should change “five genes” to “three genes” (line 243), replace “focussed” with “focused” (line 111), and run a full spell check.	We have checked the revised manuscript carefully to ensure there are no discrepancies to correct spelling mistakes.	Minor changes to text to ensure clarity and consistent spelling
	Minor 3. Separate spliced and unspliced plots obscure kinetic parallels. The authors should overlay the traces (e.g., combine Fig 1k and Fig 2a) to better illustrate matching expression dynamics.	To appreciate the kinetic parallels it is necessary to look at data from individual subjects, using “phase portraits”. We have added a new figure to illustrate this.	Added Supplementary Figure 1 showing phase portraits for IFI44L and Ly6E, and new text at Line 247: “...which is also evident in the corresponding phase portraits (Supplementary Fig. 1)”
	Minor 4. Velocity arrows in fate maps are qualitative and lack quantitative backing. The authors should provide numeric summaries and consider adding a “group to group velocity” statistic to VeloCD.	Where RNA velocity arrows are depicted on fate maps their length and direction represent the magnitude and direction of RNA velocity for each sample across all relevant genes. Therefore, we consider them to be quantitative.	We have clarified this in the text, Line 277: “Arrows on these fate maps represent the magnitude and direction of RNA velocity for each sample across all relevant genes.”

Figure for review only, assessment for batch effects

As part of normal quality control and exploratory analysis procedures we examine the effect of "batch" in low dimensional transcriptomic space (via PCA). Batch effects can occur as result of plate, extraction, cleanup, experiment and library preparation method. Using the information available to us for each of the RNA-Seq datasets, we examined these in each dataset discussed in our publication. As shown, there is no evidence of batch effect in these datasets and these various "batch" groups heavily overlap, regardless of the type of "batch" examined. The plots shown were generated using the biplot method, with the shaded areas generated using the encircle method (for each batch group). No "batch" data was available for the INSTINCT and IBD datasets.